# Shifts in pore connectivity from precipitation versus groundwater rewetting increases soil carbon loss after drought

A. Peyton Smith [1], Ben Bond-Lamberty [2], Brian W. Benscoter [3], Malak M. Tfaily [4], C. Ross Hinkle [5], Chongxuan Liu [6,7] & Vanessa L. Bailey [1]

Droughts and other extreme precipitation events are predicted to increase in intensity, duration, and extent, with uncertain implications for terrestrial carbon (C) sequestration. Soil wetting from above (precipitation) results in a characteristically different pattern of pore-filling than wetting from below (groundwater), with larger, well-connected pores filling before finer pore spaces, unlike groundwater rise in which capillary forces saturate the finest pores first. Here we demonstrate that pore-scale wetting patterns interact with antecedent soil moisture conditions to alter pore-scale, core-scale, and field-scale C dynamics. Drought legacy and wetting direction are perhaps more important determinants of short-term C mineralization than current soil moisture content in these soils. Our results highlight that microbial access to C is not solely limited by physical protection, but also by drought or wetting-induced shifts in hydrologic connectivity. We argue that models should treat soil moisture within a three-dimensional framework emphasizing hydrologic conduits for C and resource diffusion.

[1] Biological Sciences Division, Pacific Northwest National Laboratory, 902 Battelle Boulevard, Richland, WA 99352, USA. [2] Joint Global Change Research Institute, Pacific Northwest National Laboratory, 5825 University Research Court, Suite 3500, College Park, MD 20740, USA. [3] Florida Atlantic University, Department of Biological Sciences, 3200 College Avenue, Davie, FL 33314, USA. [4] Environmental Molecular Sciences Laboratory, Pacific Northwest National Laboratory, 902 Battelle Boulevard, Richland, WA 99352, USA. [5] University of Central Florida, Ecosystem Processes and Services Laboratory, 4110 Libra Drive, Orlando, FL 3216, USA. [6] Physical Sciences Division, Pacific Northwest National Laboratory, 902 Battelle Boulevard, Richland, WA, 99354, USA. [7] Southern University of Science and Technology, School of Environmental Science and Engineering, 518055 Shenzhen, China. Correspondence and requests for materials should be addressed to A.P.S. (email: apeyton.smith@gmail.com) or to V.L.B. (email: vanessa.bailey@pnnl.gov)

Climate change is altering global precipitation patterns: droughts are predicted to increase in intensity, duration, and geographic coverage, with major implications for soil carbon (C) storage at ecosystem and global scales[1–3]. Precipitation events are becoming less common but more intense in the majority of warm, humid environments in the contiguous United States[4,5]. How drought and wetting events will alter terrestrial C uptake and loss remains highly uncertain, particularly for soils, which comprise the Earth's largest terrestrial C reservoir[6,7]. Laboratory and field studies indicate drought-affected soils produce a $CO_2$ pulse when rewet[8–10], but models do a poor job of reproducing these moisture-related patterns of greenhouse gas (GHG) emissions in soils[7], limiting our ability to predict how drying and rewetting will influence soil C source or sink capacity under scenarios of altered precipitation[6].

A particular source of uncertainty concerns pore-scale soil biogeochemical processes that underpin larger-scale C flux responses to soil wetting events[11,12]. Particular pore size domains have distinct microenvironments that may favor different types of microorganisms[12,13] and mechanisms of soil C protection[14]. Soil structure, particularly the size and connectivity of soil pores, has been shown to affect microbial activities[15,16], bulk soil decomposition rates during cycles of drying and wetting[17], and organic matter (OM) complexity[18]. Wetting direction (i.e., from above via precipitation or from below via groundwater rise) produces alternate saturation patterns among different pore-size domains: for example, when groundwater rises, capillary forces first saturate the finest pores, whereas in precipitation events, gravitational forces first saturate coarse, well-connected large pores[19,20]. Because fine-sized pores are associated with more aromatic and condensed forms of OM[18], rewetting from groundwater rise may make complex forms of C more susceptible to mineralization than precipitation-driven rewetting, producing rapid core-scale to ecosystem-scale C loss.

The commonly observed pulse in $CO_2$ that occurs when dried or drought-affected soils are rewet (i.e., the Birch Effect[9]) is rarely investigated at multiple spatial or temporal scales, and its mechanistic underpinnings can vary, making predictions difficult. Recent research suggests that the Birch Effect is a dual response, driven by rapid changes to microbial biomass growth[21] and activation of extracellular enzymes[22]. For short-term droughts (<2 weeks), the bacterial growth response to rewetting is linear and immediate, whereas for longer droughts bacterial growth is exponential, though this follows a lag period that may be up to 18 h[23]. The drier the soil and longer the drought, the greater the pulse of $CO_2$ upon rewetting[23,24]. These responses are often rapid and short-lived, occurring within 24–48 h[24–28]. However, short-term responses can result in significant C losses from rewetting, because such hot moments can comprise a substantial fraction of the landscape-scale or annual flux budget[29].

The objective of this research was to develop a molecular understanding of the influence that wetting direction and antecedent soil moisture have on soil C vulnerability at both the soil pore and core-scale. Given the immediacy of the microbial responses to rewetting, we focused our measurements on the 20 h immediately following rewetting. We hypothesized greater short-term $CO_2$ emissions would be observed during bottom-wetting (simulated groundwater rise) relative to top-wetting (simulated precipitation), as C occluded in fine pores is more readily accessed through capillary diffusion-driven rewetting, and that this effect would be more pronounced in soil cores subjected to laboratory-induced drought conditions prior to wetting. We also hypothesized that the abundance of complex C compounds (such as lignin, tannin, and condensed hydrocarbons) would increase in pore water collected from drought-conditioned soils due to drought-induced changes in sorption/desorption interactions

between soil minerals and OM[30,31]. Drought-induced accumulation of dead or dormant microbial biomass could also reduce lipids in pore water. We tested this hypothesis on soil cores that were structurally intact, and for which physical protection may have been a dominant mechanism for C persistence, as well as on soil cores that had been homogenized, so that any effect of physical protection was removed.

We used a laboratory experiment to uncover short-term pore-scale and core-scale mechanisms governing the C source or sink capacity of soils in response to drought and rewetting direction. Intact soil cores were collected from a sandy site located in the Everglades watershed (FL, USA) naturally subject to significant hydrologic variability, including capillary-led wetting[32]. Sixteen experimental cores were randomly assigned to four factorial treatments of antecedent soil moisture conditions (moisture at time of sampling, vs. antecedent drought) and rewetting direction (simulated precipitation vs. groundwater rise). To capture the immediate response to rewetting, $CO_2$ and $CH_4$ flux rates were monitored during and for 20 h after wetting. Pore water was then collected from each core using different suctions to sample water retained by pore throats of different effective size domains[18] (−1.5, −15, and −50 kPa suctions representing pore throat diameters of ~200, 20, and 6 μm[33]) and characterized via ultrahigh resolution mass spectrometry. To further clarify the importance of physical protection in controlling soil C fluxes, the rewetting experiment was then immediately repeated on the same, previously intact soil cores after each core was homogenized (see Methods section, Supplementary Fig. 1). Core-scale $CO_2$ and $CH_4$ flux and pore-scale OM composition measured from homogenized cores represents the response from previously protected C that was physically occluded in intact cores. To test the degree to which these dynamics might be observed at larger scales, field-scale soil $CO_2$ emissions were analyzed using precipitation and groundwater data collected within the Disney Wilderness Preserve (DWP) (Supplementary Fig. 2).

Our results reveal that effective pore size domain is a strong predictor of both the composition and concentration of soluble C, emphasizing the importance of pore-scale (i.e., physical) protection of soil C. At the core-scale, we show that differences in short-term $CO_2$ and $CH_4$ production depends on antecedent moisture content and on the direction of soil rewetting. In situ $CO_2$ emissions are also influenced by the direction of soil wetting suggesting that precipitation and groundwater fluctuations may interact to destabilize soil C at the field scale. Our results highlight that microbial access to soil C is governed by physical proximity and hydrologic connectivity, which are sensitive to changes in soil moisture content and wetting direction.

## Results

**Pore-scale organic matter composition**. The molecular composition of OM in pore waters was correlated with effective pore-size domain, with drought and wetting direction affecting the abundance of C compounds within individual pore water fractions (Fig. 1). In a principal components analyses of compound classes[34] inferred from the FT-ICR-MS spectra, the first principal component (PCA axis 1) was solely influenced by pore water fraction ($P < 0.0001$) (Fig. 1). FT-ICR-MS features that correlated with PCA axis 1 include "unnamed" compounds (i.e., assigned peaks that contributed to the total number of C molecules detected, but did not fit into any of the eight compound classifications) (−96.7%) and lignin (75.8%) (Supplementary Table 1). This was supported by the observed relative increase in unnamed compounds ($P < 0.0001$) and relative decrease in lignin ($P < 0.0001$) and tannins ($P = 0.030$) in more loosely held pore water fraction (−1.5 kPa) compared to more tightly held pore

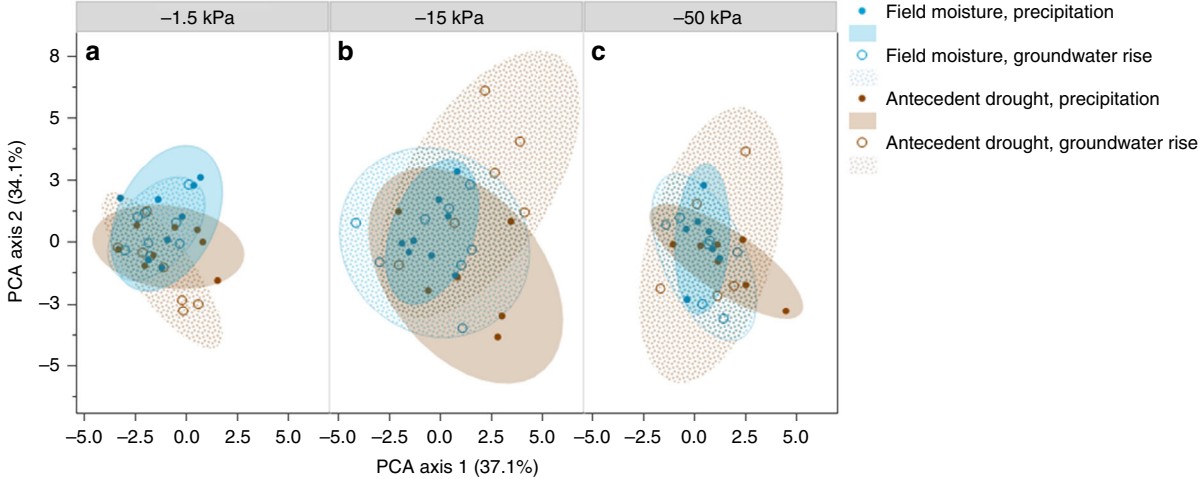

**Fig. 1** Principal components analysis using the molecular composition of water-soluble organic carbon collected from different pore size suction domains. A principal components analysis (PCA) of Fourier-transform ion cyclotron resonance (FT-ICR) mass spectrometry defined organic carbon compound classes (lipids, unsaturated hydrocarbons, lignin, proteins, and so on) for soil pore water collected at **a** −1.5 kPa suction representing coarse-sized pores and pore-throats (approx. >300 μm diameter), **b** −1.5 kPa suction representing medium-sized pores and pore-throats (approx. 20–300 μm diameter), and **c** −50 kPa suction representing fine pores and pore-throats (approx. 6–20 μm diameter) collected from soil cores subjected to antecedent drought or maintained at field moisture conditions and to different rewetting directions (rewet from the top to simulate precipitation wetting, or from below to simulate capillary-led groundwater rise) in the laboratory. Blue shaded points and solid-shaded 90% confidence interval ellipse represent pore water collected from soil cores subjected to field moisture and precipitation-led rewetting conditions ($n = 8$, 8, and 7 for −1.5, −15, and −50 kPa pore water fractions, respectively), whereas open blue circles and pattern-filled confidence intervals represent field moisture core rewet from below to simulate groundwater rise ($n = 8$, 8, and 7). Brown shaded points and the corresponding solid-shaded 90% confidence interval ellipse represent pore water collected from soil cores subjected to antecedent drought and rewet via simulated precipitation ($n = 8$, 6, and 7), whereas open brown circles and pattern-filled confidence intervals represent data from drought-conditioned samples rewet via simulated groundwater rise ($n = 7$, 6, and 5). Correlation coefficients associated with the PCA are included in Supplementary Table 1

waters (−50 kPa) (Fig. 2, Supplementary Table 2). We also observed a relative enrichment in unsaturated hydrocarbons in the more loosely held pore water fraction compared to the intermediate-suction pore water fraction (−15 kPa, $P = 0.039$).

Soil homogenization influenced the second principal component (Fig. 1, PCA axis 2, $P = 0.042$), with significant interactive effects between wetting direction and effective pore size domain ($P = 0.013$), and drought and wetting direction ($P = 0.039$). Proteins and lipids were highly correlated (80.9, 80.9%, respectively) with PCA axis 2 (Supplementary Table 1). The sole effect of soil homogenization on the relative abundance of compound classes was observed as a depletion of unnamed compounds in loosely held pore waters ($P = 0.033$) (Fig. 2).

When pore water fractions from intact and homogenized cores were analyzed separately, antecedent drought decreased the relative abundance of lipids in the loosely held pore water fraction (−1.5 kPa, $P = 0.047$) and increased the relative abundance of carbohydrates in intermediately held pore water (−15 kPa, $P = 0.022$) in intact cores (Table 1, Supplementary Table 2). In intermediately held pore water, the relative abundance of carbohydrates also increased in cores rewet via groundwater rise compared to precipitation rewetting ($P = 0.032$). In the tightly held pore water fraction (−50 kPa), antecedent drought followed by precipitation-led rewetting resulted in a relative enrichment of lignin ($P = 0.013$) and relative depletion of unnamed compounds ($P = 0.045$) compared to cores rewet via groundwater rise and also compared to core maintained at field moisture conditions.

In homogenized soils, antecedent drought also decreased the relative abundance of lipids in the loosely held pore water fraction (−1.5 kPa, $P = 0.009$). (Table 1, Supplementary Table 2). The loosely held pore water fraction was also relatively depleted in lipids ($P = 0.034$), proteins ($P = 0.0495$), and carbohydrates ($P = 0.031$) in cores rewet via groundwater rise vs. precipitation. Tannins

were relatively enriched in both the loosely held ($P = 0.024$) and intermediately held (−15 kPa, $P = 0.032$) pore water fractions collected from homogenized, drought-conditioned cores compared to cores maintained at field moisture conditions. In addition, we observed significant interactive effects of antecedent drought and rewetting direction on the relative abundance of lipids ($P = 0.018$), condensed hydrocarbons (0.029), and carbohydrates ($P = 0.021$) in the intermediately held pore waters. Similar interactions were observed for unsaturated hydrocarbons in tightly held pore water (−50 kPa). As a result, the effect of rewetting direction was only observed in drought-conditioned soils.

An indicator of OM molecular richness, i.e., the total number of C features (m/z peaks) identified using FT-ICR-MS, increased with soil homogenization ($P = 0.001$), antecedent drought ($P = 0.011$), and simulated groundwater rise ($P = 0.034$) (Fig. 3). Effective pore size domain did not have an effect on total peaks identified, except in intermediately held pore waters collected from soil cores subjected to drought ($P = 0.049$), where peaks increased. When intact and homogenized cores were analyzed separately, antecedent drought resulted in more FT-ICR-MS peaks in all pore water from intact cores ($P = 0.0497$), whereas in homogenized cores antecedent drought increased the number of peaks only in intermediately held pore water (−15 kPa) ($P = 0.0377$). Due to technical limitations with low volume samples (Methods section), the total concentration of organic carbon and nitrogen in pore waters was highly variable (Supplementary Table 3). The greatest variability and the highest values of water-soluble organic carbon (WSOC) were observed in pore water collected from homogenized cores that were originally pre-conditioned to drought (Supplementary Table 3). The total amount of pore water collected was not influenced by antecedent soil moisture conditions or rewetting direction for intact and homogenized cores, but did differ among pore water suctions

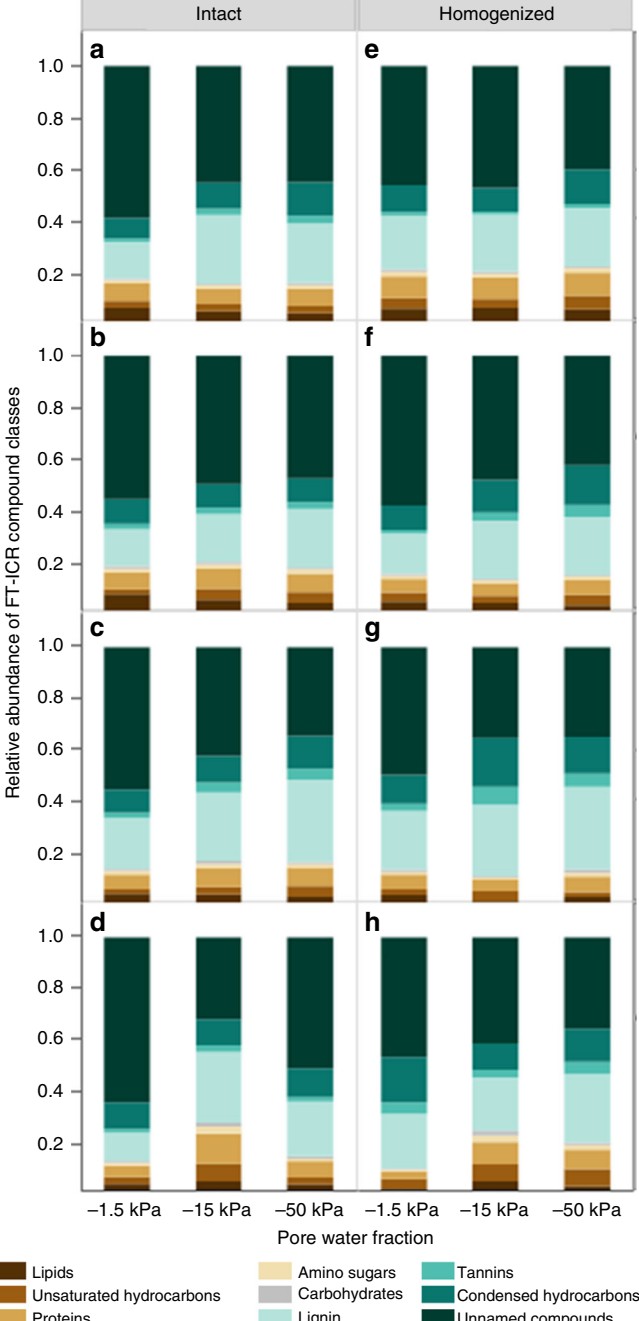

**Fig. 2** Pore-scale molecular composition of water-soluble organic carbon collected after rewetting incubation from different pore size suction domains. The relative abundance of Fourier-transform ion cyclotron resonance (FT-ICR) mass spectrometry defined organic carbon compound classes (lipids, unsaturated hydrocarbons, lignin, proteins, and so on) for soil pore water collected at −1.5, −15, and −50 kPa suctions for intact cores maintained at field moisture conditions and rewet via **a** simulated precipitation ($n = 4$, 4 and 4 for −1.5, −15, and −50 kPa pore water fractions, respectively), **b** simulated groundwater rise ($n = 4$, 4, and 3), and from cores subjected to antecedent drought and rewet via **c** simulated precipitation ($n = 4$, 3, and 4), **d** simulated groundwater rise ($n = 4$, 3, and 2), or from homogenized soil cores maintained at field moisture conditions and rewet via **e** simulated precipitation ($n = 4$, 4, and 3), **f** simulated groundwater rise ($n = 4$, 4, and 4), and from cores subjected to antecedent drought and rewet via **g** simulated precipitation ($n = 4$, 3, and 3), **h** simulated groundwater rise ($n = 3$, 3, and 3). Soil pore water was collected immediately following rewetting and post-rewetting incubation. Statistical summaries for Fig. 2 are included as Table 1. Mean and standard error values for compound classes shown are included in Supplementary Table 3

with more pore water collected at the lowest suction (−1.5 kPa), representing the coarsest pore size domain (pores restricted by size and pore-throat diameters >300 μm) (Supplementary Table 4). The effect of soil homogenization on the pore size distribution was minimal (Supplementary Fig. 3) with greater frequency (~25%) of pores 150–200 μm diameter in homogenized cores compared to intact cores (~17%) and a greater overall diversity of pore sizes in intact cores compared to homogenized cores (Supplementary Movie 1).

**Core-scale gas fluxes**. Fluxes for both $CO_2$ (Fig. 4a, b) and $CH_4$ (Fig. 4c, d) were strongly affected by antecedent drought ($P = 0.0002$) and soil homogenization ($P < 0.0001$) with multiple significant interactive effects between different combinations of drought, wetting direction, and soil homogenization. When intact and homogenized cores were analyzed separately, there was a significant interactive effect of rewetting direction and antecedent drought on cumulative $CO_2$−C and $CH_4$−C in intact cores, whereas there were no significant treatment effects on cumulative $CO_2$−C and $CH_4$−C in homogenized cores (Table 2). In intact cores, antecedent drought and precipitation rewetting (wet from above) resulted in the greatest amount of cumulative $CO_2$−C ($P = 0.010$), whereas antecedent drought and groundwater rise (rewetting from below) resulted in the greatest cumulative $CH_4$−C ($P = 0.021$) (Table 2). More specifically, in intact cores, antecedent drought followed by precipitation rewetting emitted 4.7 times more cumulative $CO_2$−C than precipitation-rewet field moisture cores, 2.5 times more than field moisture cores rewet via groundwater rise, and 1.5 times more than drought-conditioned cores rewet via groundwater rise (Table 3). Antecedent drought followed by rewetting via groundwater rise resulted in 8.8 times more cumulative $CH_4$−C compared to cores maintained at field moisture content and rewet via groundwater rise, but there was no difference in cumulative $CH_4$−C in cores rewet via precipitation regardless of antecedent soil moisture (i.e., drought-conditioned or field moisture) for intact soil cores (Tables 2, 3). In sharp contrast to intact soil cores, homogenized cores exhibited no significant effect of antecedent drought, wetting direction or their interaction on either gas emissions (Table 2).

Cumulative $CO_2$−C was positively correlated with the amount of water imbibed ($r = 0.76$, $P = 0.001$, $n = 16$) in intact cores, regardless of wetting direction. Prior to rewetting, moisture contents (both gravimetric and volumetric) were negatively correlated with cumulative $CO_2$−C in these cores ($r = 0.50$, $P = 0.045$ for both, $n = 16$); no correlations were observed with moisture after rewetting. This is consistent with differences we observed in pre-rewetting and post-rewetting moisture contents by treatment (Supplementary Table 5): intact soil cores subjected to antecedent drought conditions had significantly lower pre-wetting moisture contents compared to soil cores maintained at field moisture conditions, but there was no difference in their moisture contents immediately following rewetting between field moisture and antecedent drought cores (Supplementary Tables 5, 6).

**Field-scale gas fluxes**. Field observations of $CO_2$ emissions following precipitation events and groundwater fluctuations were generally consistent with the pore-scale and core-scale effects observed in the laboratory. Soil wetting, whether by groundwater rise or precipitation, significantly influenced soil $CO_2$ emissions at the field scale (Fig. 5). Despite a weak relationship ($r = 0.44$), the amount of precipitation ($P = 0.040$) and the interaction between precipitation and groundwater elevation ($P = 0.005$) significantly altered in situ $CO_2$ flux, with increased $CO_2$ emissions after precipitation-led soil wetting.

**Table 1 *P*-values for the main and interactive effects of antecedent drought and wetting direction for molecular compound classes of individual pore water domains collected from intact and homogenized cores**

| Effects | Pore water suction | Lipids | Unsaturated hydrocarbons | Condensed hydrocarbons | Proteins[a] | Amino sugars | Carbohydrates[a] | Lignin | Tannins[a] | Unnamed compounds |
|---|---|---|---|---|---|---|---|---|---|---|
| *Intact* | | | | | | | | | | |
| Drought | −1.5 kPa | **0.047** | NS | NS | NS | NS | NS | NS | NS | NS |
| Wetting Direction | | NS | NS | NS | NS | NS | NS | NS | NS | NS |
| Drought x Wetting | | NS | NS | NS | NS | NS | NS | NS | NS | NS |
| Drought | −15 kPa | NS | NS | NS | NS | NS | **0.022** | NS | NS | NS |
| Wetting Direction | | NS | NS | NS | NS | NS | **0.032** | NS | NS | NS |
| Drought x Wetting | | NS | NS | NS | NS | NS | NS | NS | NS | NS |
| Drought | −50 kPa | NS | NS | NS | NS | NS | NS | NS | NS | NS |
| Wetting Direction | | NS | NS | NS | NS | NS | NS | **0.014** | NS | **0.012** |
| Drought x Wetting | | NS | NS | NS | NS | NS | NS | **0.013** | NS | **0.045** |
| *Homogenized* | | | | | | | | | | |
| Drought | −1.5 kPa | **0.009** | NS | **0.011** | NS | NS | NS | NS | **0.024** | NS |
| Wetting Direction | | **0.034** | NS | NS | **0.0495** | NS | **0.031** | NS | NS | NS |
| Drought x Wetting | | NS | NS | **0.022** | NS | NS | NS | NS | NS | NS |
| Drought | −15 kPa | NS | NS | NS | NS | NS | NS | NS | **0.032** | NS |
| Wetting Direction | | NS | NS | NS | NS | NS | NS | NS | NS | NS |
| Drought x Wetting | | **0.018** | NS | **0.029** | **0.0496** | NS | **0.021** | NS | NS | NS |
| Drought | −50 kPa | NS | NS | NS | NS | NS | NS | NS | NS | NS |
| Wetting Direction | | NS | NS | NS | NS | NS | NS | NS | NS | NS |
| Drought x Wetting | | NS | **0.027** | NS | NS | NS | NS | NS | NS | NS |

*P*-values marked in bold are considered significant $P < 0.05$, non-significant *P*-values are reported as NS
Residual maximum likelihood model (REML) results with relative abundance of Fourier-transform ion cyclotron resonance (FT-ICR) mass spectrometry defined organic carbon compound classes (lipids, unsaturated hydrocarbons, lignin, proteins, and so on) as response variables for soil pore water collected at −1.5, −15, and −50 kPa suctions. FT-ICR compounds; proteins, carbohydrates and tannins were log-transformed for normality in REML tests
[a]log transformed for normality

## Discussion

Effective pore size domain was a stronger predictor of both the composition and concentration of soluble C in pore water than antecedent drought or rewetting direction. More tightly held pore waters (−50 kPa), sampled from isolated, fine (~6 μm diameter) pores and pore-throats, contained relatively more 'complex' C compounds (e.g., lignin and tannin-like compounds) compared to the more loosely held pore waters (e.g., −1.5 kPa, coarse-sized, more connected pores, >200 μm). These results are consistent with previous results measured from similarly located soils at DWP, Florida, where soluble-OM associated with more tightly held pore water was composed of relatively more lignin, tannins, and condensed hydrocarbons, whereas more loosely held pore water was more enriched in lipids[18].

Antecedent soil moisture (i.e., drought), however, altered C processes at multiple scales[11,35]. At the pore-scale, antecedent drought showed a decrease in lipids in loosely held pore water, possibly indicating a loss of microbial biomass due to prolonged dry conditions[36], as the cellular membrane of microorganisms is dominated by phospholipids[37,38]. Conversely, pore water tannins and lignin increased following drought. This is consistent with our hypothesis that the abundance of complex C compounds (such as lignin, tannin, and condensed hydrocarbons) would increase in pore water collected from soil subjected to antecedent drought. The relative increase in complex C compounds may be due to a negative enrichment from preferential degradation of other compounds (such as lipids, from above), or due to the physio-chemical relationship between ionic strength and the sorption of C to mineral surfaces. When soils are subjected to drying conditions, the ionic strength increases, resulting in the release of C compounds that were previously sorbed to mineral surfaces[30,31]. This results in a greater solubilization of previously protected, complex (e.g., high molecular weight) C compounds[31].

At the core-scale, the higher $CO_2$ and $CH_4$ fluxes from drought-conditioned soil relative to the field moist cores during rewetting was consistent with the Birch Effect[9] and may have been due to a rapid release of microbial osmolytes from the sudden shift in soil water content[35,39,40]. It may have also been due to increased hydrologic connectivity collocating decomposers with previously inaccessible C and/or microbial necromass-C that accumulated during drying[36]. As we hypothesized, we detected fewer lipids in the loosely held pore water sampled from drought-conditioned soil compared to cores maintained at field moisture, suggesting accumulated microbial residues was rapidly mineralized when drought-conditioned soils were rewet, leading to higher $CO_2$ and $CH_4$ emissions. While the majority of studies report that the pulse in mineralization is often short-lived, returning to respective field moist emissions within hours or days[8], our study highlights that differences in the immediate response for $CO_2$ and $CH_4$ production may depend on the direction of soil rewetting.

Even during our short-term rewetting experiment, we observed a legacy effect from antecedent drought on $CO_2$ rates for intact soil cores during the post-wetting incubation (Fig. 4a). In our study, drought-conditioned soils imbibed and held more water upon rewetting than cores maintained at field moist conditions (Supplementary Table 5). At the same time, the moisture content of all intact soil cores (both those maintained at field moisture and antecedent drought conditions) did not differ after being rewet. Thus, drought legacy and wetting direction (precipitation or groundwater rise) are perhaps more important determinants of short-term C mineralization than current soil moisture content in these soils. Current models and representations of soil C dynamics are based on a 'snapshot' of current soil conditions, with no consideration of history[35,41]. Taking historical patterns of drought and precipitation into account may improve such representations[42,43].

While we hypothesized greater $CO_2$ emissions from drought-stricken soils rewet from below (groundwater rise), simulated precipitation actually emitted more $CO_2$ from intact soils. This suggests that while our assumptions about the spatial distribution of soluble C were correct, we did not take into account how the

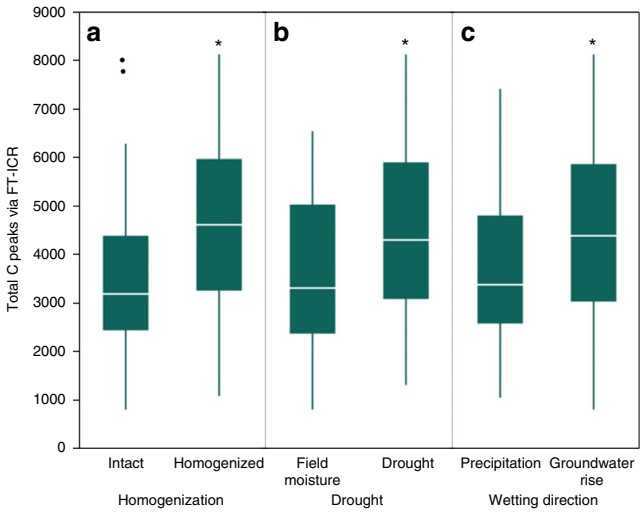

**Fig. 3** Total C peaks identified in soil pore waters by soil homogenization, antecedent drought and wetting direction. The total number of Fourier-transform ion cyclotron resonance (FT-ICR) mass spectrometry peaks of organic C identified across all pore water fractions that significantly differed by **a** soil homogenization ($P = 0.001$, $n = 43$ for intact, 42 for homogenized), **b** antecedent drought ($P = 0.011$ l, $n = 46$ for field moisture, 39 for antecedent drought), and **c** wetting direction ($P = 0.034$, $n = 44$ for simulated precipitation, 41 for simulated groundwater rise). The outlier box plot whiskers represent the first and third quartile minus or plus, respectively, 1.5 times the interquartile range. Soil pore water was collected immediately following rewetting and post-rewetting incubation

vertical distribution of C would affect $CO_2$ production. The mean bulk C content for field moist soil cores (0–15 cm soil depth) was $0.64 \pm 0.10\%$ C (Supplementary Table 5). However, greater amounts of C were located in the top 3 cm of the cores ($1.0 \pm 0.01\%$ C compared to $0.1 \pm 0.001\%$ C at bottom, data not shown). Thus, soils wet via simulated precipitation experienced rapid hydrologic connectivity between decomposers and greater amounts of C than soils wet via simulated groundwater rise resulting in more C mineralization and $CO_2$ production. In addition, coarser and more connected pores, pores that precipitation-led wetting would fill first, may contain more microbial biomass than finer, more isolated pores thus leading to greater respiration[12,13,37]. The increase in cumulative $CH_4$–C in drought-conditioned cores compared to cores maintained at field moisture only when cores were wetted from below may be driven more by the functional potential of the microbial to produce $CH_4$ than the concentration of C. For example, there may be a higher abundance of microorganisms functionally capable of producing $CH_4$ at the bottom of the core, where $O_2$ concentrations are less abundant, compared to the top of the core[44].

The functional potential of decomposers may also explain why we did not observe our hypothesized increase in $CO_2$ emitted from soil wet via groundwater rise. Capillary-led wetting created a hydrological conduit that connected fine-pore C with decomposers; however, those decomposers may not have been capable of degrading the forms of C associated with finer pores (lignin-like or tannin-like compounds). For example, fungi and other known lignin and tannin degraders may not be able to physically access C located in the finest pore domains that are most isolated from the hydrologically connected pore network[45]. Accordingly, competent decomposers need more than a hydrologic conduit in order to access C; they also need physical access, highlighting the importance of both hydrologic and physical accessibility in determining the vulnerability and persistence of C in soil.

Sandy soils, such as the ones in this study, are often characterized as having a greater abundance of coarse-sized pores relative to fine pores and thus not often recognized for strong structural C protection mechanisms. Structural mechanisms of C protection in soils are often attributed to soils with high aggregate stability, low aggregate turnover, and in soils with a greater proportion of microaggregates and micropores[14]. The process of homogenization distributes C, resources and decomposers more equally throughout the core, removing both the vertical and pore-size controls on C mineralization observed in our results from intact soil cores. However, despite the sandy nature of these soils, more soluble C was collected from homogenized soil cores compared to intact cores, suggesting that there was a pool of structurally protected C that we were unable to access and identify when we first collected pore water from intact cores.

Changes in soil structure (e.g., pore architecture) and pore-scale moisture conditions ultimately affect microbial activities and greenhouse gas emissions[15,16,46]. For example, Negassa et al.[13] showed that a greater abundance of water-filled pores <100 µm diameter led to greater $CO_2$ emissions than from larger pores (100–2000 µm diameter). Kravchenko and Guber[47] demonstrated that greater C losses occurred in pores ~30–90 µm in diameter and for greater protection of C via adsorption to minerals and physical barriers in pores <5 µm, which can even extend all the way to pores <30 µm[17]. Franzleubbers[46] shows that soils with coarser-sized pore distributions can have equal cumulative C mineralization loss as soils with a greater abundance of finer-sized pores depending on the proportion of water-filled pore space. Drought can reduce C mineralization in large, well-connected pores first compared to fine pores as water availability and substrate diffusion is limited, resulting in a greater accumulation of organic matter[15,16,45]. At the same time, reduced oxygen availability in fine pores may also limit microbial activity and $CO_2$ production[48]. Therefore, it appears that these 3D structures and hydrologic conduits for substrate-diffusion and resource-diffusion strongly control the metabolic potential of the soil[37]. It is evident that C dynamics occurring at the core-scale, which is often the scale at which we measure $CO_2$ and C concentrations, are ultimately driven by processes occurring at the pore-scale. Incorporating such pore-scale processes may enhance model accuracy[47].

At the field scale, we are unaware of any studies showing differential effects of rain vs. groundwater inputs on the soil-to-atmosphere $CO_2$ flux. Depth to water table has been occasionally shown to be important for understanding soil $CO_2$ emissions[49–51], but generally soil moisture is used in measurement and modeling analyses[52] without any regard as to the how water moves through soils (e.g., precipitation or groundwater rise). We found that the in situ $CO_2$ flux responded to both precipitation events and to fluctuations in groundwater level. While this is correlative and observational, and thus not conclusive proof, it is consistent with our laboratory results and supports the idea that soil wetting direction can be a strong control on field-scale $CO_2$ emissions as well. Given that soil respiration is the second-largest C flux at ecosystem to global scales, and a generally poorly constrained one[53], our results may help generate new hypotheses that could be posed to understand larger-scale C fluxes. Furthermore, the combined analyses of our laboratory experiment with the field experiment suggest that there is the potential to improve current predictions of soil C dynamics by including soil history and wetting source in future models.

Our results highlight that accessibility is not solely determined by physical proximity, but also by hydrologic connectivity. Our results support current theories that the soil-pore matrix plays a profound role in the distribution of C, decomposers and the fate

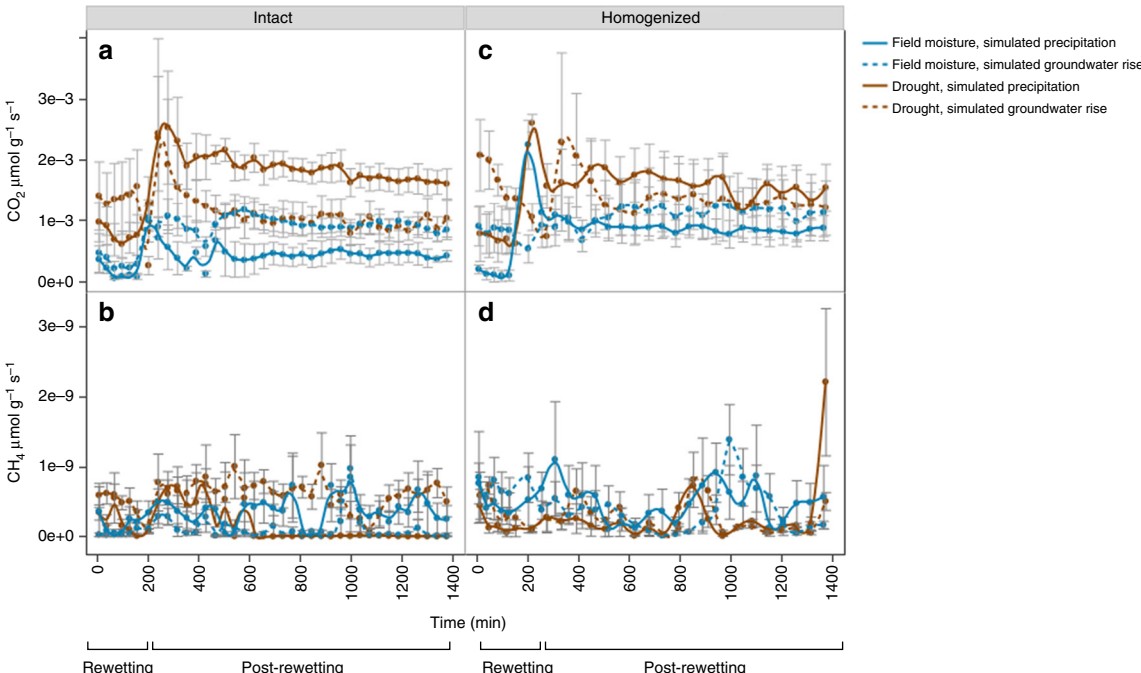

**Fig. 4** Core-scale carbonaceous greenhouse gas measurements throughout rewetting and post-rewetting incubation. $CO_2$ ($\mu$mol g$^{-1}$ s$^{-1}$) respiration rates from **a** intact or **b** homogenized cores, and $CH_4$ ($\mu$mol g$^{-1}$ s$^{-1}$) respiration rates from **c** intact or **d** homogenized cores, during rewetting (200 min) and 20 h following rewetting event; wet from above (simulated precipitation) or wet from below (simulated groundwater rise) for cores maintained at field moisture content or subjected to laboratory-induced antecedent drought cores. Solid lines represent cores from simulated precipitation, whereas dashed lines represent cores from simulated groundwater rise treatments. Bars represent standard error for $n = 4$ cores, with the exception of homogenized cores subjected to antecedent drought and wet from below where $n = 3$. Statistical summaries for cumulative $CO_2$−C and $CH_4$−C are included as Table 2

of OM transformations, and that microbial access to C is a dominant mechanism controlling the persistence of C in soils. The core-scale lab research revealed a strong role for pore-scale protection of soil C, with the potential for this C to relocate differently depending on the wetting direction and on exposure to antecedent drought. While our laboratory experiment cannot accurately represent in situ phenomena, our findings are consistent with the field-scale observations, suggesting that precipitation and groundwater fluctuations may interact to destabilize soil C at the field scale. This suggest that ecosystem C models need to treat soil moisture not as a single number, but within a 3D framework emphasizing hydrologic conduits for substrate and resource diffusion[19]. It is difficult to expect that current model uncertainties[7,54] will be resolved, or even effectively constrained, without significant efforts in these areas. As droughts and shifts in precipitation patterns increase with climate change, understanding how drought and precipitation events interact at a variety of scales is essential in order to improve predictions of the C sink/source capability of soils.

## Methods

**Field sampling**. Intact soil cores (3 cm diameter, 15 cm height) were sampled in September 2014 from a pine flatwoods stand (28.104641°, −81.419027°) in DWP near Kissimmee, FL, USA (Supplementary Fig. 2). DWP has a humid, subtropical climate with a mean annual temperature of 22.4 °C and precipitation of 1222 mm yr$^{-1}$. Soils are dominated by sandy textures, and depending on local topographic position show moderate to high levels of SOM accumulation at the surface. The soil is classified as an Immokalee fine sand. The Immokalee series is taxonomically defined as sandy, siliceous, hyperthermic Arenic Alaquods, and is characterized as being poorly drained, friable, strongly acidic, with a weak fine granular structure due to the mixture of organic matter and fine roots[55]. Groups of 16 cores (4 × 4) were sampled from 0.25 × 0.25 m square areas; four sets of such cores were taken from locations 2 m apart, for a total of 64 cores sampled from 0 to 15 cm depth. For this study, 16 cores were randomly selected out of the 64 cores collected. Soil cores were stored at −20 °C for 48 h per USDA-APHIS requirements, and then shipped to the laboratory overnight on blue ice.

**Experimental design**. A full factorial design was used to test the effects of soil moisture (field moist vs. laboratory-induced drought) and wetting direction (simulated precipitation vs. simulated groundwater rise) on core-scale carbonaceous greenhouse gas production and pore-scale C chemistry (Supplementary Fig. 1). The cores were randomly divided into these four treatment combinations: field moist core wet from above, field moist core wet from below, drought core wet from above, and drought core wet from below. Soil moisture manipulations occurred as a laboratory pre-treatment. $CO_2$ and $CH_4$ were measured from all soil cores during rewetting (~4 h) and for a short-term incubation post-rewetting (~20 h, total incubation of ~24 h). Immediately thereafter, soil pore water was collected and characterized (see below). The soil inside each core was then removed, homogenized using a 2 mm sieve (U.S. Standard Testing Sieve, Advantech Manufacturing, New Berlin, WI, USA), and repacked to nearly an identical bulk density as the original soil core. Roots and rocks were removed prior to repacking the homogenized soil. In order to measure core-scale fluxes and pore-scale OM chemistry associated with the pool of previously occluded (i.e., physically protected) C, the repacked, homogenized soil cores were immediately rerun through the experiment, starting with the rewetting incubation (Supplementary Fig. 1). A more detailed description of the soil moisture pretreatments, core-scale and pore-scale measurements follows.

**Soil treatments**. Soils were subjected to one of two pretreatments: maintained at field moisture content or subjected to a laboratory simulated drought. All intact soil cores (PVC tubing, AMS, Inc. American Falls, ID, USA) were fit with 100 μm mesh screen at the base to maintain core integrity. Field moisture cores were maintained at their original, in situ moisture content by weight, ~15%. Drought was simulated by placing a set of cores on a dry ceramic pressure plate (1 bar Tempe Pressure Cell units, Soil Moisture Equipment Corp. Goleta, CA, USA) and allowing them to evaporate until they reached ~5% moisture by weight in an environmental growth chamber set at 22 °C and 60% humidity (~30 days) (BBDW80, Conviron, Winnipeg, Manitoba, Canada).

There were two wetting treatments, both referred to as "rewetting", designed to simulate precipitation or groundwater rise. The amount of water received by each soil core during rewetting was controlled so that each wetting direction treatment (from above to simulate precipitation or from below to simulate groundwater rise) experienced similar rates of rewetting. We used a rate of wetting that was calculated based on preliminary measurements of natural imbibition (used here as the process of taking in water by the soil core) on a set of soil cores ($n = 6$) not used in the remainder of this study. To simulate groundwater rise, soil cores were placed on a saturated ceramic pore plate and allowed to naturally imbibe water for 200 min (based on preliminary observations of when cores stopped imbibition). For the

**Table 2 Statistical summary for cumulative carbonaceous greenhouse gas emissions from incubated cores measured during rewetting and 20 h post rewetting**

| Variable | Effect | Intact cores | | | Homogenized cores | | |
|---|---|---|---|---|---|---|---|
| | | df | F ratio | *P*-value | df | F ratio | *P*-value |
| CO₂−C | | | | | | | |
| | Drought | 1 | 22.597 | **<0.0001** | 1 | 4.490 | NS |
| | Wetting direction | 1 | 0.647 | NS | 1 | 0.002 | NS |
| | Drought x Wetting direction | 1 | 9.170 | **0.010** | 1 | 0.038 | NS |
| CH₄−C | | | | | | | |
| | Drought | 1 | 3.609 | NS | 1 | 0.018 | NS |
| | Wetting direction | 1 | 0.095 | NS | 1 | 0.0004 | NS |
| | Drought x Wetting direction | 1 | 7.043 | **0.021** | 1 | 0.020 | NS |

REML models performed on non-transformed CO₂−C data and log-transformed CH₄−C data
*P*-values marked in bold are considered significant <0.05, non-significant *P*-values are reported as NS
Statistical summary including degrees of freedom (df), Fisher test statistic (F ratio), and the calculated probability (*P*-value) derived from residual maximum likelihood mixed effects models (REML) testing the effects of antecedent drought, wetting direction and the interaction of drought and rewetting (Drought x Rewetting Direction) on cumulative carbon emissions from CO₂ and CH₄ respired during rewetting and for 20 h post-rewetting for intact and homogenized cores

simulated precipitation treatment, cores were rewet from above using a peristaltic pump (Cole Palmer, Vernon Hills, IL, USA) for 30 min at 0.53 ml min$^{-1}$, followed by 0.082 ml min$^{-1}$ for 170 min (again, based on preliminary observations of natural imbibition rates) in order to control for and reproduce similar rates of wetting to that of soils wet from below. Despite our best attempts at controlling the rate and amount of water imbibed, there were differences in the amount of water imbibed among cores and treatments. We also observed that the initial wetting front was more rapid, but less uniform, in cores rewet via simulated precipitation compared to groundwater rise (i.e., water rapidly percolated to the bottom of the core without complete horizontal saturation). By the end of rewetting (200 min), all cores appeared to have equal distributions of water throughout the soil core regardless of wetting direction or antecedent soil moisture conditions. See Supplementary Table 5 for the amounts of water imbibed averaged across treatments. The amount of water imbibed was measured as the difference in weight for each soil core and was measured immediately following the rewetting treatment (i.e., after 200 min).

**Core-scale measurements**. Carbonaceous greenhouse gas (CO₂ and CH₄) concentrations in the cores' headspace were measured using a G2301 Picarro GHG analyzer (Picarro, Sunnyvale, CA, USA) during the wetting period (which lasted for 200 min) and after it (up to 20 h) to capture the immediate response of the soil to wetting. Fluxes were computed from the concentration changes according to the following Eq. (1):

$$A = \left(\frac{dC}{dt} \frac{V}{M} \frac{Pa}{RT}\right), \qquad (1)$$

where $A$ is the flux (μmol g soil$^{-1}$ s$^{-1}$), $dC/dt$ the rate of change in gas concentration (mole fraction s$^{-1}$), $V$ the total chamber volume (cm$^3$), $M$ dry soil mass (g), $Pa$ atmospheric pressure (kPa), $R$ the universal gas constant (8.3 × 10$^3$ cm$^3$ kPa mol$^{-1}$ K$^{-1}$), and $T$ air temperature (K). The raw data and the R processing code can be found at https://github.com/bpbond/dwp_peyton. When both core-scale and pore-scale (as described below) analyses were completed, soil cores were broken apart and a variety of soil physical and chemical properties were measured, including soil mass, bulk density, and total C and nitrogen (N). Bulk density was calculated for each core as the mass of air-dried soil divided by the volume of the intact soil core. Subsamples of homogenized, air-dried soil were ground to <250 μm using a ceramic mortar and pestle and sent to the University of Wisconsin-Madison for total C and N, where triplicates were randomly arranged and run on a Flash 2000 NC analyzer (Thermo Scientific, Wilmington, DE, USA) combustion elemental analyzer. For porosity measurements and pore size distributions, an intact and homogenized soil core was scanned using X-ray Computed Tomography (XCT) on an X-Tek/Metris XTH 320/225 kV scanner (Nikon Metrology, Belmont, CA). Data were collected at 110 kV and 265 μA X-ray power. The core samples were rotated continuously during the scans with momentary stops to collect each projection (shuttling mode) while minimizing ring artifacts. A total of 3142 projections were collected over 360° with 0.5 s exposure time and 4 frames per projection. Image voxel size was 28 microns. The images were reconstructed to obtain 3D data sets using CT Pro 3D (Metris XT 2.2, Nikon Metrology). Representative slice and 3D images were created using VG Studio MAX 2.1 (Volume Graphics GmbH, Heidelberg Germany). Image processing and porosity analysis (including pore volume segmentation and pore analysis) was carried out using ImageJ 1.51k (National Institute of Health, USA).

**Pore-scale measurements**. Pore waters were sampled by transferring each core onto individual 100 kPa Tempe Pressure Cell units to sequentially collect pore waters at −1.5, −15, and −50 kPa using a dual valve pressure controller (Alicat

Scientific, Tucson, AZ, USA), novel method modified from Lentz[56]. Suction strengths were chosen based on preliminary work showing these particular pore water fractions are discrete for these soils[18]. Using the Kelvin equation[33] to estimate the largest water-filled pore diameter, −1.5, −15, and −50 kPa suctions correspond to pore and pore-neck size diameters of ∼200, 20, and 6 μm[33]. It is important to note that water collected at each suction better represents water contained within soil pore spaces restricted by channels, or pore throats, rather than held within pores of approximated diameters. For clarity, these three pore water fractions (200, 20, and 6 μm) will be referred to according to their suction strength or collectively referred to as "effective pore size domains". Pore water was pulled for 24 h, starting with the lowest strength, and stored at −20 °C until further analysis. Because little to no pore water was collected at higher suctions (−15 and 50 kPa), replication was reduced from $n = 4$ to $n = 3$ or 2 (depending on the treatment) for select pore water analyses. As such, the number of replicates used in each analysis is included in each table or Figure caption. For low volume pore water samples, we prioritized characterizing the molecular composition of C in pore water rather than measuring the concentration of water-soluble N (WSN) and organic C (WSOC), due to technical limitations in measuring low volume samples for WSN and WSOC. Concentrations of total C (water-soluble organic carbon, WSOC) and N (WSN) in pore water were determined via combustion catalytic oxidation (TOC-5000A TOC analyzer, Shimadzu, Columbia, MD, USA). The molecular composition of the C dissolved in the pore water was characterized by electrospray ionization (ESI) coupled with Fourier-transform ion cyclotron resonance mass spectrometry (FT-ICR). The samples were desalted by solid phase extraction (SPE) with PPL cartridges following Dittmar et al.[57]. Samples were first acidified to a pH of 2 before extraction, and the water-soluble organic matter eluted in methanol; see Tfaily et al.[58]. The extracts were then injected directly on a 12 Tesla Bruker SolariX FT-ICR spectrometer. A standard Bruker ESI source was used to generate negatively charged molecular ions.

Samples were then introduced to the ESI source equipped with a fused silica tube (200 μm i.d) through a syringe pump at a flow rate of 3.0 μL min$^{-1}$. Experimental conditions were as follows: needle voltage, +4.4 kV; Q1 set to 150 $m/z$; and the heated resistively coated glass capillary operated at 180 °C. Ninety-six individual scans were averaged for each sample and internally calibrated using organic matter (OM) homologous series separated by 14 Da (−CH₂ groups). The mass measurement accuracy was less than 1 p.p.m. for singly charged ions across a broad $m/z$ range (i.e., 200, <$m/z$ <1200). Chemical formulas were assigned using in-house software based on the Compound Identification Algorithm described by Kujawinski and Behn[59] and modified by Minor et al.[60]. Chemical formulas were assigned based on the following criteria: signal-to-noise (S/N)>7, and mass measurement error <1 p.p.m., taking into consideration the presence of C, H, O, N, S, and P and excluding other elements.

FT-ICR spectra were classified into eight biomolecular groups, referred to as FT-ICR compound classes, based on O/C and H/C counts; lipids (0 < O/C ≤ 0.3, 1.5 ≤ H/C ≤ 2.5), unsaturated hydrocarbons (0 ≤ O/C ≤ 0.125, 0.8 ≤ H/C < 2.5), proteins (0.3 < O/C ≤ 0.55, 1.5 ≤ H/C ≤ 2.3), amino sugars (0.55 < O/C ≤ 0.7, 1.5 ≤ H/C ≤ 2.2), carbohydrates (0.7 < O/C ≤ 1.5, 1.5 ≤ H/C ≤ 2.5), lignin (0.125 < O/C ≤ 0.65, 0.8 ≤ H/C < 1.5), tannins (0.65 < O/C ≤ 1.1, 0.8 ≤ H/C < 1.5), and condensed hydrocarbons (0 ≤ O/C ≤ 0.95, 0.2 ≤ H/C < 0.8)[34,61] (Supplementary Fig. 4). FT-ICR compound classes are tentative classifications as they are solely based on the O/C and H/C ratios from the molecular formula, not the molecular structural. As such, it would be more accurate to describe compounds as lipid-like or carbohydrate-like. For simplicity, we will refer to each compound class, however, as lipids, tannins, proteins, and so on. Relative abundance values were calculated from count values associated with each observed biomolecule group normalized by the total number of C molecules identified. FT-ICR spectra that did not fit into any of the eight biomolecule group classifications, but that contributed to the total

**Table 3 Cumulative carbonaceous greenhouse gas emissions from incubated cores measured during rewetting and for 20 h post rewetting**

| | Intact cores | | | Homogenized cores | | |
|---|---|---|---|---|---|---|
| | n | CO$_2$–C (mg) | CH$_4$–C (µg) | n | CO$_2$–C (mg) | CH$_4$–C (µg) |
| Field moisture | | | | | | |
| Simulated precipitation | 4 | 51.7 ± 19.8 **c** | 0.0536 ± 0.026 **ab** | 4 | 108.6 ± 28.2 | 0.0785 ± 0.051 |
| Simulated groundwater rise | 4 | 100.2 ± 4.1 **bc** | 0.0117 ± 0.005 **b** | 4 | 118.7 ± 45.0 | 0.0551 ± 0.016 |
| Antecedent drought | | | | | | |
| Simulated precipitation | 4 | 245.1 ± 17.4 **a** | 0.0288 ± 0.013 **ab** | 4 | 208.6 ± 41.7 | 0.0546 ± 0.019 |
| Simulated groundwater rise | 4 | 161.2 ± 34.5 **b** | 0.1032 ± 0.030 **a** | 3 | 202.0 ± 59.4 | 0.0418 ± 0.008 |

Letters not shared among rows show that means are significantly different using Student's Least Significant Means *t*-test
Mean and standard error values for cumulative CO$_2$–C (mg), and CH$_4$–C (mg) from intact and homogenized soil cores maintained at field moisture content, or subjected to laboratory-induced antecedent drought for both wetting directions (i.e., wet from above to simulate precipitation or wet from below to simulate groundwater rise).

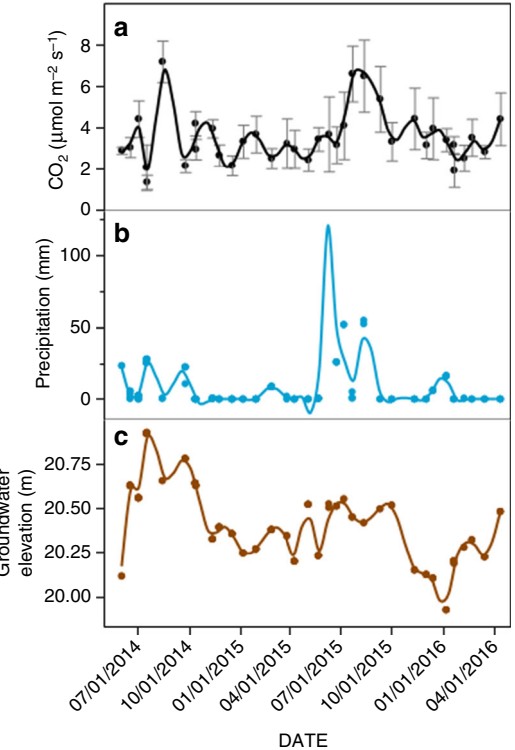

**Fig. 5** Field-scale carbon dioxide flux, precipitation and groundwater rise. Field measurements of **a** mean daily CO$_2$ flux (µmol m$^{-2}$ s$^{-1}$) respiration (bars represent standard error of $n = 8$ plots), and weather tower reports at Walker Ranch Weather Station at the Disney Wilderness Preserve (FL) of **b** total precipitation (mm) and **c** mean groundwater level (m) for 24 h prior to CO$_2$ measurements. Using a residual maximum likelihood mixed effects model, CO$_2$ emissions were influenced by precipitation ($P = 0.040$) and by the interaction of precipitation and groundwater elevation ($P = 0.005$). Measurements were made from 2 June 2014 to 13 April 2016, for a total of 156 measurements

number of C molecules detected, were reported as unnamed compounds. The processing code can be found at: https://github.com/ktoddbrown/FTICR_Processing.

**Field-scale measurement.** We examined the abiotic drivers of field-measured CO$_2$ emissions using data collected from a series of eight 1 m$^3$ closed chambers at DWP (28.105466˚, −81.415755˚), (Supplementary Fig. 2) measured roughly monthly from 31 August 2013 to 13 April 2016 (total $n = 156$). The soil is classified as a Smyrna sand, of which the series is taxonomically defined as sandy, siliceous, hyperthermic Aeric Alaquods. This series is characterized as poorly drained, fri-able, strongly acidic, rapidly permeable at the surface and having a weak coarse

granular structure[55]. Soil respiration was measured under dark conditions (0 µmol m$^{-2}$ s$^{-1}$ photosynthetically active radiation), established by placing an opaque shroud over the closed chamber. CO$_2$ measured from chambers may also include respiration from existing understory vegetation consists; wiregrass (*Aristida stricta*) and palmetto (*Serenoa repens*). Headspace CO$_2$ concentration and temperature were measured at 1.6 s intervals using an EGM-4 IRGA (PP Systems, Amesbury, MA) equipped with the TRP-2 air temperature/PAR probe and the headspace air recirculated over a 5 min incubation period. Chamber volume was corrected for small variation in soil surface elevation within the chamber using an elevation model (10 cm grid resolution) generated for each chamber and the slope of CO$_2$ exchange over incubation time was used to calculate the flux rate (positive values indicate increased headspace concentration). There was a small subset of CO$_2$ data collected in 2013, but the measurements were infrequent and we chose to work with a subset of data starting when measurements were taken more regularly (approx. monthly) from 14 March 2014 to 13 April 2016 (total $n = 156$). Precipitation data recorded from a nearby weather station (WRWX, 28.04872777˚, −81.3998305˚) and groundwater elevation data from a well (WR9 + GW1, 28.0632051˚, −81.2509225˚) were also used for the same dates CO$_2$ measurements were made. Both weather and groundwater stations are maintained by the South Florida Water Management District (SFWMD); publically available data were downloaded from DBHYDRO: http://my.sfwmd.gov/dbhydroplsql/show_dbkey_info.main_menu.

**Statistical and data analysis.** Residual maximum likelihood (REML) models, a mixed-effect approach, were used to identify main and interactive effects of antecedent drought, wetting direction and soil homogenization on core-scale CO$_2$ and CH$_4$ measurements. In order to account for potential biased variance and covariance estimates between laboratory replicates, which were intact soil cores that varied in volume and other parameters due to field-based heterogeneity, the soil core (from which the measurement was taken from) was included as a random effect. The number of replicates used in each analysis differed due to experimental constraints (e.g., not enough pore water collected at specific suctions or for specific treatments) and is included in each table or figure caption for clarification. Core-scale CO$_2$ and CH$_4$ emissions (µmol g$^{-1}$ s$^{-1}$) were transformed for normality using a reciprocal log transformation. Cumulative CO$_2$–C and CH$_4$–C (log-transformed), were also analyzed using REML models. Due to significant main and interactive effects of soil homogenization potentially confounding other results, the effects of antecedent drought and wetting direction were testing individually for intact or homogenized cores in flux measurements.

Pore-scale REML models used relative abundance values of FT-ICR compounds classes (lipids, carbohydrates, lignin, and so on) as response variables. In addition to the effects of antecedent drought, wetting direction, and soil homogenization, pore water fraction (based on different suction strengths soil water was collected at; −1.5, −15, and −50 kPa) was also included as a fixed effect. Due to the significant main and interactive effects of pore water fraction on all FT-ICR compound classes, the effects of antecedent drought, wetting direction and soil homogenization was tested for each pore water fraction individually. Intact and homogenized core data were considered paired samples.

Field-scale, or in situ CO$_2$ fluxes, groundwater elevation and precipitation were also analyzed using REML models with each CO$_2$ chamber considered a random variable. The response variable (CO$_2$ respiration) was somewhat non-normally distributed, but log-transforming did not significantly affect the results. All statistical analyses were performed using JMP Pro Version 13 (SAS Inst. Inc., Cary, NC, USA).

**Data availability.** All data from this study have been deposited on Figshare—DOI: 10.6084/m9.figshare.5349082.

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

## Acknowledgements

This research is based on grants to both the Pacific Northwest National Laboratory (PNNL) and to the University of Central Florida (UCF) from the U.S. Department of Energy, Office of Science, Biological and Environmental Research as part of the Terrestrial Ecosystem Sciences Program. PNNL is operated for DOE by Battelle Memorial Institute under contract DE-AC05-76RL01830. UCF funds were awarded via grant DE-SC00008301. A portion of this research was performed using EMSL, a DOE Office of Science user facility sponsored by the DOE's Office of Biological and Environmental Research and located at Pacific Northwest National Laboratory. We thank those who helped in this research: Drs Tamas Varga, Erika Marín-Spiotta, Li-Jung Kuo, Kenton Rod, and Kathe Todd-Brown.

## Author contributions

This study was conceived by V.L.B., who also performed the field sampling. V.L.B. and A.P.S. designed the study. A.P.S. led the laboratory analyses with M.M.T. and C.L. assisting with specialized instrumentation. C.R.H. and B.W.B. led the field-scale flux measurements. A.P.S. and B.B.-L. analyzed data, and A.P.S. wrote the manuscript with significant contributions from all authors.

## Additional information

**Competing interests:** The authors declare no competing financial interests.

