## [Peer Review File · Nature Communications]

Reviewers' comments:

Reviewer #1 (Remarks to the Author):

A. Summary of the key results

Direction (or possibly pattern of) re-wetting influences the magnitude of the respiratory flush of CO₂ or CH₄ realized in a sandy soil

Originality and interest: if not novel, please give references

The general subject (improving our understanding and ability to model moisture X mineralization interactions) is important but the study methods and soil selected for use may not provide meaningful mechanistic insight into riparian soils that may receive increasing amounts of their moisture through bottom up processes. The authors need to strengthen their case, bolster the data shared, and improve the discussion of results to convince me that this would shed new light on this topic. It is interesting that the authors do not cite earlier works that characterize the effects of rewetting on C availability and moisture flush. Ultimately one would want to know if bottom wetting after drought increased mineralization because zones of greater wetness coincided with areas of more abundant substrate. Are differences in types of compounds reflective of the intensity of wetting effects (is this linked to the classical understanding that drying kills microbes) or to solubilization or desorption (due to greater moisture) or, are lignin-like compounds or lipids a product of drying/condensation due to legacy of drought? I hope these points can be clarified.

B. Data & methodology: validity of approach, quality of data, quality of presentation

I appreciate that the authors are trying to assess a complex system where processes are changing in space and time even in the short duration of the incubation. This is why more information about moisture treatments needs to be provided to help determine whether moisture conditions (soil spatial domains accessed) were really similar or different. I believe that more water was added to the re-wetting treatments, would have moved faster- possibly created zones of greater moisture content and filled larger pore domains- would have assumed that capillary rise might move through smaller pores but possibly not if rate for top watering slow enough. Was water dripped on to soil surface directly and was there a free water surface? Were the moisture and temperature at the top of the cores and bottom of the cores the same? Do moisture and C substrate measures taken at the end of the incubation represent conditions present during flux measurement? Change in rates over time might provide some insight into that?

Fig 1 and 2 present treatments in different order. Colors are consistent but it would be easier to follow if both presented field moist then drought

C. Appropriate use of statistics and treatment of uncertainties

I had some concerns about interpretation and presentation of statistical results. Explanations for some of the statistical analyses need to be improved (noted on manuscript). In some instances, data appear to be over interpreted- For example, 'Extended Data Table 1. Core-scale physical and chemical properties' reports mean differences for drought by wetting in several instances where, according to the footer legend, there was not a significant interaction. The main effect was drought and this was not clearly reported. The inherent variability shown in the table suggests that some of the results might be affected by core heterogeneity.

D. Conclusions: robustness, validity, reliability

While I agree with the main thesis of the paper, which is that spatially-complex moisture dynamics must be considered by models, the paper did not add to my understanding of those factors.

Discussion identified results but did not interpret them in an additive or mechanistic manner. The authors may possess some additional data that could shed light on mechanisms and allow them to make more conclusive use of their results. As written, conclusions were made in nebulous terms- results of various sorts were not brought together to provide a unified understanding and lab and field methods were only very vaguely associated.

Rather than simply noting fluctuations in water-table level and precipitation observed in the field 'had an interactive effect on in situ CO₂ production' could the authors note what those interactions were and link them in a logical manner to results produced by the core the incubations?

E. Suggested improvements: experiments, data for possible revision

I encourage the authors to expand the legends for tables and figures so that information can be more easily related to results. For example, Extended Data Table 1. Core-scale physical and chemical properties does not reveal exactly when the data were collected. I suspect this was immediately following the incubation. Also, n=16 suggests that these might be from a subset of cores.. which ones?

Extended Data Table 3. Suggests that temperature effects or warming was a major factor - (by far the largest F values). Means for temperature should be added to the table and appropriate discussion added. This made me wonder about temperature in the cores.

Moisture release data should be summarized and included in the manuscript. This would provide essential info on the size of pores that were rewet. It would have been nice to have a secondary measure of moisture that might reveal spatial heterogeneity- were cores evenly wet in both top and bottom watered cores? Did wetting fronts move at significantly different rates?

It would be helpful to see the anova for fig 2 to make sure it fairly represents data set. Data summary that averages contents across all treatments by pore sizes makes me wonder if means meet criteria for fisher protected means comparisons.. if I understand the fig legend and text, this is the result of statistically significant drought X wetting direction X pore domain interaction? From the looks of it, coarse pores had no significant interaction, drought effect increased abundance of lipids. For carbs, seems like comparison maybe should consider all classes ? And, for fine pore classes, were results based on the same number of cores or were some with no drainage?

References: appropriate credit to previous work?

Maybe include some note of classical C flush interpretation and acknowledge the limited pore size range to be found in the sandy soil? . Wang, W J, et al. "Relationships of soil respiration to microbial biomass, substrate availability and clay content." *Soil biology & biochemistry* 35.2 (2003):273-284. ..

F. Clarity and context: lucidity of abstract/summary, appropriateness of abstract, introduction and conclusions

Am reprinting my comments on abstract here.. For this to move forward I would hope for an abstract that explained or postulated why bottom up wetting increased losses from CO₂ and CH₄. Would like the summary to jointly address treatment effects on substrate abundance, and moisture constrains (how does C type inform us about physical access?) - They may be able to exploit differences between CO₂ and CH₄ results to note that enhanced loss of CH₄ would signal greater amount of area experiencing anaerobic conditions. As presented, results are somewhat intriguing but not informative.

to show that wetting direction (simulated precipitation versus groundwater rise) significantly changes pore-scale water C chemistry and core-scale greenhouse gas emissions. We found that wetting direction and soil moisture history (field moisture versus drought) significantly altered both core-scale greenhouse gas (CO₂ and CH₄) production and pore-scale C chemistry . Antecedent drought increased cumulative CO₂-C and CH₄-C loss 4.7 times with simulated precipitation and 3.1 times with simulated groundwater rise compared to field moist soil cores under simulated precipitation. At the pore-scale, soluble C was more complex in fine-sized pores compared to coarse-sized pores, with drought and wetting direction influencing the relative abundance of lipids, carbohydrates, and lignin only in coarse, medium and fine-sized pores, respectively. At the field scale, fluctuations in water-table level and precipitation had an interactive effect on in situ CO₂ production.

Reviewer #2 (Remarks to the Author):

This ms reports the impact of two different rewetting sources on CO₂ and CH₄ production of a previously dry riparian soil under laboratory and field conditions. Both, simulated precipitation and groundwater rise affected the CO₂ and CH₄ production as well as the relative abundance of C compounds in soil pore water. The fact that capillary rise of groundwater into dry soil horizons can trigger an increase in microbial processes has been widely overlooked in soil C research. However,

the ms has some methodological shortcomings and provides little mechanistic investigations that together allow no generalization of the findings:

(1) Soil cores (0-15 cm depth) were taken from one study site with subtropical climate and then stored at -20°C . It seems that the topsoil in the field is not influenced by capillary rise (Fig. 3c, groundwater level varies between -2 and -3m), but the soil at about <-1.5 m. Soil freezing at -20°C could have altered the microbial community, which never experienced soil frost, and the amount of available C through killed microbial cells. Taking soil cores (core diameter?) further interrupts hyphae of mycorrhizal and saprotrophic fungi and roots that have the ability to redistribute water in soils along soil water potential gradients.

(2) The origin of CO_2 and CH_4 is unclear. Isotopic analyses could help to resolve the sources. In particular, changes in the ground water level could lead to a short-term volatilization of gases in addition to changes in microbial gas production. Consumption of CH_4 is another factor controlling the net CH_4 efflux.

(3) Drought stress was induced by reduction to $\sim 5\%$ soil moisture. Water potential would have been more appropriate considering variations in soil texture. How is physiological drought stress defined? Is 5% soil moisture light, moderate or strong drought stress? Small changes in water content can have tremendous effects on the physiological relevant water potential.

(4) Oxygen concentration in soil pores is a crucial parameter for the balance of CH_4 and CO_2 production which might have been affected by the direction of wetting. Are data available?

(5) The rationale for sampling of soil pore water by pressure cells at -1.5, -15 and -50 kPa is not clear to me. Were sampled water volumes different for specific pore sizes and treatments? Relative composition of organic solutes (Fig. 2, Ex-Table 2) is less relevant for CO_2 and CH_4 fluxes than total amount of organic solutes.

L60-61: 'Groundwater-led wetting affects the vast majority of riparian soils, which hold roughly a third of the global, terrestrial carbon pool.' Do you mean wetlands? Riparian soils, generally defined as soils along rivers and lakes, make a minor contribution to global wetland area. I don't see the global significance of drying/wetting in riparian soils for the global C cycle.

Reviewer #3 (Remarks to the Author):

This is a review of the manuscript "Precipitation versus groundwater: rewetting source affects soil carbon loss after drought". The authors used laboratory incubations of intact soil cores to induce antecedent drought conditions (with appropriate controls), and rewetting of cores from either the top down (simulating precipitation) or the bottom up (simulating groundwater rise). They found that wetting direction and moisture history altered carbon dioxide and methane fluxes from the cores, as well as pore-scale carbon compound distribution. The authors claim that drought and wetting effects on pore-scale C distribution and hydrological connectivity interact to alter core-scale greenhouse gas fluxes with implications at the field scale.

The claims made are novel in that few studies have examined the effects of precipitation and groundwater rise together in the same experiment. It is interesting to see, and makes intuitive sense that precipitation primarily affects carbon dioxide flux and groundwater rise affects methane flux. The data on pore scale carbon-compound distribution are particularly interesting and will probably be the most interesting to other researchers in this area. The results are somewhat difficult to interpret, but the pattern is pretty clear. However, I do think that the authors may have overstated the claim that their data show "that accessibility to microbial degradation is a dominant protect mechanism of C in these soils" (lines 130-131). While their data are consistent with this hypothesis, it is not clear whether changes in compound abundance (lipids in this case) are due to microbial mortality, decomposition, mass transport of microbes and substrate, or even if the isolated lipids are derived from microbial or plant sources. All of these are mentioned in the manuscript, but the data available cannot separate the various fate and transformation mechanisms. I think it would make a better paper if the authors were to back off this claim and

discuss in a little more detail each potential process, and how the net effect of these processes might explain the observed patterns.

I'm also not sure how much the field-based data add to the claims made the authors (paragraph, lines 116-123). I understand that the authors want to tie their core-scale experiments to plot- and field-scale dynamics, but I have some problems with this analysis. First, These data look like they come from a separate field study in that the methods mention open top chambers placed on the soils to elevated temperatures, yet no temperature treatment was included in the lab incubation experiment. The regression analyses on field carbon dioxide fluxes also include this temperature treatment, possibly confounding the results related to moisture differences. If the analysis were limited to ambient temperature treatments only, this might make for a better comparison between laboratory and field studies. Second, the laboratory results show that precipitation has the strongest effect on carbon dioxide flux and groundwater rise on methane flux, yet the field results, which have carbon dioxide as the response variable, are really only testing the control of precipitation on field-scale carbon flux. It seems that methane flux data at the field scale are really needed to test the combined effects of wetting direction at the field-scale. I think that the manuscript is strong enough to stand on the laboratory results alone and the authors should reconsider how, or even if, these field data should be included.

Response to the Reviewers' comments

Reviewer #1 (Remarks to the Author):

A. Summary of the key results

Direction (or possibly pattern of) re-wetting influences the magnitude of the respiratory flush of CO₂ or CH₄ realized in a sandy soil

Originality and interest: The general subject (improving our understanding and ability to model moisture X mineralization interactions) is important but the study methods and soil selected for use may not provide meaningful mechanistic insight into riparian soils that may receive increasing amounts of their moisture through bottom up processes. The authors need to strengthen their case; bolster the data shared, and improve the discussion of results to convince me that this would shed new light on this topic. It is interesting that the authors do not cite earlier works that characterize the effects of rewetting on C availability and moisture flush. Ultimately one would want to know if bottom wetting after drought increased mineralization because zones of greater wetness coincided with areas of more abundant substrate. Are differences in types of compounds reflective of the intensity of wetting effects (is this linked to the classical understanding that drying kills microbes) or to solubilization or desorption (due to greater moisture) or, are lignin-like compounds or lipids a product of drying/condensation due to legacy of drought? I hope these points can be clarified.

We significantly revised the manuscript in order to address these concerns; we expanded on the methods used, strengthened our case by including more data in our analyses, cited more pivotal works on rewetting, and provided mechanistic insight in the discussion. In our response below, you will find responses to both the concerns you listed in text and the ones you list here.

B. Data & methodology: validity of approach, quality of data, quality of presentation

I appreciate that the authors are trying to assess a complex system where processes are changing in space and time even in the short duration of the incubation. This is why more information about moisture treatments needs to be provided to help determine whether moisture conditions (soil spatial domains accessed) were really similar or different.

These are excellent and relevant points to consider. We provided significantly more information about the moisture and wetting treatments in the methods, main text and in our extended data figures to clarify the potential spatial domains accessed. Specific answers to your questions and concerns are addressed below:

I believe that more water was added to the re-wetting treatments, would have moved faster- possibly created zones of greater moisture content and filled larger pore domains- would have assumed that capillary rise might move through smaller pores but possibly not if rate for top watering slow enough

Soils that experienced antecedent drought, did take in more water. But that doesn't necessarily mean they were provided more water. Every core was designed to receive the same amount of water at the same rate. Moisture contents for both sets of cores (field moisture and drought-conditioned) were equal following wetting (200 minutes), see lines 429 - 442: *"The amount of water received by each soil core during rewetting was controlled so that each wetting direction treatment (from above to simulate precipitation or from below to simulate groundwater rise) experienced similar rates of rewetting...For the simulated precipitation treatment, cores were rewet from above ... for 30 minutes at 0.53 ml/min, followed by 0.082 ml/min for 170 minutes (again, based on preliminary observations of natural imbibition rates) in order to control for and reproduce similar rates of wetting to that of soils wet from below. Despite our best attempts at controlling the rate and amount of water imbibed, there were differences in the amount of water imbibed among cores and treatments"*

Wetting would naturally be faster in the cores subjected to antecedent drought, however, even in cores maintained at field moisture conditions, water was naturally imbibed quite rapidly - with the greatest rate of rewetting occurring during the first 30 minutes (see rate above), learned from preliminary experiments.

Was water dripped on to soil surface directly and was there a free water surface?

Water was directly dripped onto the surface and rapidly percolated through the core – there was no standing water on the top of the core.

Were the moisture and temperature at the top of the cores and bottom of the cores the same?

At the end of rewetting (200 minutes), water appeared to be consistently distributed throughout the core. At the start of wetting, the moisture content at the top and bottom would differ depending on which wetting treatment (top or bottom) the core received.

The cores experimented on were small cores (3 cm diameter and 15 cm tall) (reported in line 387) and were all experimented on at 22°C (reported in line 426) using water that was also maintained at 22°C. As such, we can only assume that the temperature of the top and bottom core were equal.

Do moisture and C substrate measures taken at the end of the incubation represent conditions present during flux measurement?

Moisture contents were measured immediately before and after (200 minutes) rewetting and after the post-rewetting incubation (at 24 hours). The post-rewetting incubation moisture contents were equal to moisture contents measured at 200 minutes (i.e. immediately following the rewetting event). Therefore, we can assume that moisture contents from 200-1400 minutes were constant (i.e. post-rewetting incubation).

Change in rates over time might provide some insight into that?

We provided the change in rate (of CO₂ and CH₄ emissions) over time, as requested (see Figure 4). We were unable to provide hourly changes in soil C and soluble C during such as short-term incubation, but since soluble C was extracted immediately following the 24 hour incubation and both the moisture contents and fluxes stabilized following rewetting, we can assume that the soluble C measured represents conditions present during the flux measurements, most accurately during flux measurements from 200- 1400 mins.

Fig 1 and 2 present treatments in different order. Colors are consistent but it would be easier to follow if both presented field moist then drought

We have revised our figures and made sure that colors not only remain consistent, but represent the same treatment organization.

In-text comments:

A20: What was core diameter...only 15 cm?

Core diameter included. (reported in line 387)

A21: Were wetting treatments the same or different for the field moist and drought treatments. it seems like you adjusted exposure to have the field moist treatments take up similar quantity of water? Is that the case? Details about this are critical to interpreting results.

See response above. (detailed in lines 429 – 442)

A22: So was water dripped on like rainfall or was this rate determined to create a head of water? We need to know what pores water was likely to move through.

Water dripped like rainfall at rate determined from preliminary experiments on natural imbibition in soil cores collected at the same time and at the same site. (See lines 436 - 440)

A23 Results related to relative pore domains occupied should be reported in main text- this is a critical piece of information needed to interpret results

Pore domains reported in text, see lines 100 - 102: *“Pore water was then collected from each core using different suctions to sample water retained by pore throats of different effective size domains²⁵ (-1.5, -15 and -50 kPa suctions representing pore throat diameters of approx. 200 μm, 20 μm, and 6 μm³⁰)”*

A24: This isn't clear, did they drop cores that had no flow? Move to stats and explain where this matters. I assume this is for DOC composition.

We clarified which methods and analysis had missing replicates in the methods and in the figures, tables. See lines 479 - 481: *“Because little to no pore water was collected at higher suctions (-15 and 50 kPa), replication was reduced from n = 4 to n = 3 or 2 (depending on the treatment) for select pore water analyses. As such, the number of replicates used in each analysis is included in each table or figure caption.”* and lines 548 - 550: *“The number of replicates used in each analysis differed due to experimental constraints (e.g. not enough pore water collected at specific suctions or for specific treatments) and is therefore included in each table or figure caption for clarification”*. If there was not enough pore water collected at the highest suction (-50 kPa) necessary for a particular measurement (such as DOC concentration or composition) for a core, we did not remove all other measurements taken from that core, as those other pore water fraction were relevant to the study.

C. Appropriate use of statistics and treatment of uncertainties

I had some concerns about interpretation and presentation of statistical results. Explanations for some of the statistical analyses need to be improved (noted on manuscript). In some instances, data appear to be over interpreted- For example, 'Extended Data Table 1. Core-scale physical and chemical properties' reports mean differences for drought by wetting in several instances where, according to the footer legend, there was not a significant interaction. The main effect was drought and this was not clearly reported. The inherent variability shown in the table suggests that some of the results might be affected by core heterogeneity.

We revised and clarified our results and statistics performed in the text and figures. For example, we included corresponding statistical summary tables for the tables detailing mean and standard error summaries. You can see the heterogeneity between cores as standard error values in Extended Data Table 2 and 3 (DOC composition and concentration, respectively). We also included the estimated component variances for cores (included in our REML models as a random effect) in table 2.

In-text comments:

A25: In addition to abundance we need to know what the volume of solution obtained from different pore domains- legend should highlight the fact that these data were taken post incubation- pre and post-incubation results would be needed to draw solid conclusions about how wetting altered the relative supply and subsequent loss of various C molecules

We included the volume of solution obtained from different pore size domains in the results and in Extended Data Table 3. We also included when pore water was collected and measured in the methods and in all the figure legends as requested. While we attempted to collect and measure pore water prior to rewetting in these soils (as you suggested), it was not successful. There was not enough pore water collected in field moist soils (and there was no water collected in drought-induced soils) to measure the concentration and composition of DOC pre-wetting.

A26: Seemed like text suggested n=2 for soln obtained from fine pores?

All replicates are now included in figure legends and/or in tables.

A27: Were measures taken immediately after incubation so you assume that these represent conditions experienced by microbes that contribute to respiratory flush?

This has been clarified in figure legend and in methods. See responses above.

A28: Were these from drought treatments only ?

This table has been revised into a figure (Figure 2) and includes data from all cores.

A29: If interaction not significant then Ismeans should not be broken out to evaluate the interaction effect

This has been revised – see responses above.

A30: So these don't differ among pore classes so why does figure select medium size to emphasize?

This has been revised and clarified. Statistical analysis of FT-ICR compounds in pore waters was performed on individual pore water fractions- see lines 560 - 563: *“Due to the significant main and interactive effects of pore water fraction on all FT-ICR compound classes, the effects of antecedent drought, wetting direction and soil homogenization was tested for each pore water fraction individually. Intact and homogenized core data were considered paired samples.”*

D. Conclusions: robustness, validity, reliability

While I agree with the main thesis of the paper, which is that spatially-complex moisture dynamics must be considered by models, the paper did not add to my understanding of those factors. Discussion identified results but did not interpret them in an additive or mechanistic manner. The authors may possess some additional data that could shed light on mechanisms and allow them to make more conclusive use of their results. As written, conclusions were made in nebulous terms- results of various sorts were not brought together to provide a unified understanding and lab and field methods were only very vaguely associated.

We agree that the original manuscript lacked mechanistic insight or an in-depth interpretation of the results. We significantly revised the manuscript by providing more a detailed methods and results section and by including mechanistic, or process-based, explanations of our results in the discussion section. We also added more data to the study in order to shed light on those mechanisms and improved on the ill-defined conclusions presented in the original manuscript.

Rather than simply noting fluctuations in water-table level and precipitation observed in the field 'had an interactive effect on in situ CO₂ production' could the authors note what those interactions were and link them in a logical manner to results produced by the core the incubations?

We revised the results and discussion of the field scale data, see lines 207 - 210: *“Despite a weak relationship ($r = 0.44$), the amount of precipitation ($p = 0.040$) and the interaction between precipitation and groundwater elevation ($p = 0.005$) significantly altered in situ CO₂ flux, with fluxes being higher after precipitation-led soil wetting.”* Please also see our response to the in text comment A17 below.

In-text comments:

A13: How do you sort solubilization from simple dilution or concentration that would occur with simple wetting and drying?

This has been removed.

A14: Isn't this true for all compounds in all soils?

This has been removed. Furthermore, we agree with you – accessibility is important for all compounds in soils However, a clear understanding of what compounds are accessible or protected under different climate conditions is needed.

A15: If finer pores are most isolated then there should be protection of lipids shouldn't there? Could differences be due to diffusion rather than decay?

Great questions! The majority of lipids from fresh litter inputs often enter coarse pores first, where they can be rapidly mineralized before moving to other pore spaces (see line 231). As such, they are

not usually abundant in the fine, isolated pore spaces. Microbial biomass, another important contribution to the lipid pool in soils, is also often more abundant in coarse pores compared to fine pores (see lines 230 - 232). Occluded spaces in soils, such as fine pores or microaggregates contained within macroaggregates, usually contain more microbially processed substrates or substrates, such as lignin, that lignin-degraders are unable to access due to size limitations. (see lines 316 - 318)

We believe diffusion can be just as important as decay when it comes to the distribution of C in soils. If differences were primarily due to OC diffusion alone, we would expect to see even distributions of C compounds in all pores. Reactive transport modeling is only just beginning to address the challenges of linking water movement with C access. A recent pore-scale model (developed in our lab; Yan et al 2017), used simulations to support the observations that “the soil respiration rate is controlled by the OC bioavailability under low saturation conditions and the delivery of oxygen limit under high saturation conditions.” Therefore, if diffusion were driving the differences, that would include diffusion of O₂, which would enhance decomposition.

A16: This isn't contested by anyone

We revised this statement for clarification, see lines 369 - 373: *“Our results support current theories that the soil-pore matrix plays a profound role in the distribution of C, decomposers and the fate of OM transformations, and that microbial access to C is a dominant mechanism controlling the persistence of C in soils. Our results add on to this theory, by highlighting that accessibility is not solely determined by physical proximity, but also by hydrologic connectivity”*

A17: I don't think the study is well enough controlled to determine whether water direction matters.

The field-based results were not intended to be direct evidence that wetting direction matters. The availability of the field experiment, and its relevance to our experiment provide a relevant context for our laboratory research. As such, we clarified the significance and relevance of the field-based results in the context of our study, see lines 357 - 366: *“We found that the in situ CO₂ flux responded to both precipitation events and to fluctuations in groundwater level. While this is correlative and observational, and thus not conclusive proof, it is consistent with our laboratory results and supports the idea that soil wetting direction can be a strong control on field-scale CO₂ emissions as well. Given that soil respiration is the second-largest C flux at ecosystem to global scales, and a generally poorly-constrained one⁵⁴, our results may help generate new hypotheses that could be posed to understand larger-scale C fluxes. Furthermore, the combined analyses of our laboratory experiment with the field experiment suggest that there is the potential to improve current predictions of soil C dynamics by including soil history, and wetting source in future models”*

A18: How.. what is the simple conclusion.. other than these factors interact in complex ways..

This has been removed.

A19: Numerous studies have shown this short term flush, which disappears quickly. These 24 hr incubations accentuate differences in rates.

This has now been noted in the text, see lines 270 - 273: *“While the majority of studies report that the pulse in mineralization is short-lived, returning to respective field moist emissions within hours or days⁷, our study highlights that differences in mineralization rates may depend on the direction of soil rewetting.”*

E. Suggested improvements: experiments, data for possible revision

I encourage the authors to expand the legends for tables and figures so that **information can be more easily related to results**. For example, Extended Data Table 1. Core-scale physical and chemical properties does not reveal exactly when the data were collected. I suspect this was immediately following the incubation. Also, n=16 suggests that these might be from a subset of cores.. which ones?

This is an excellent suggestion. We have revised the table and figure legends by including more details relevant to the results, including when the data was collected and how many replicates were used for

each analysis, see lines 394 – 397: *“Groups of 16 cores (4x4) were sampled from 0.25 m x 0.25 m square areas; 4 sets of such cores were taken from locations 2 m apart, for a total of 64 cores sampled from 0 - 15 cm depth. For this study, 16 cores were randomly selected out of the 64 cores collected”*

Extended Data Table 3. Suggests that temperature effects or warming was a major factor - (by far the largest F values). Means for temperature should be added to the table and appropriate discussion added. This made me wonder about temperature in the cores.

Originally, we chose to incorporate additional field data to increase our sampling size, data that included a warming treatment. However, due to concerns and confusion from several reviewers, we have decided to remove the warming plot data from our analysis of the field-scale fluxes. For the lab experiments, all cores were maintained at 22°C during the experiment and this is reported in the methods.

Moisture release data should be summarized and included in the manuscript. This would provide essential info on the size of pores that were rewet. It would have been nice to have a secondary measure of moisture that might reveal spatial heterogeneity- were cores evenly wet in both top and bottom watered cores? Did wetting fronts move at significantly different rates?

We included moisture release data in the manuscript, see Extended Data Figure 3. We are currently collaborating with modelers and performing computed x-ray tomography on similar soil cores in order to better understand pore-filling. In regards to the wetting front, see lines 193 - 197: *“We also observed that the initial wetting front was more rapid, but less uniform, in cores rewet via simulated precipitation compared to groundwater rise (i.e. water rapidly percolated to the bottom of the core without complete horizontal saturation). By the end of rewetting (200 minutes), all cores regardless of wetting direction or antecedent soil moisture conditions appeared to have equal distributions of water throughout the soil core.”*

It would be helpful to see the anova for fig 2 to make sure it fairly represents data set. Data summary that averages contents across all treatments by pore sizes makes me wonder if means meet criteria for fisher protected means comparisons.. if I understand the fig legend and text, this is the result of statistically significant drought X wetting direction X pore domain interaction? From the looks of it, coarse pores had no significant interaction, drought effect increased abundance of lipids. For carbs, seems like comparison maybe should consider all classes? And, for fine pore classes, were results based on the same number of cores or were some with no drainage?

We provided statistical summaries tables for many of the statistics reported.

References: appropriate credit to previous work? Maybe include some note of classical C flush interpretation and acknowledge the limited pore size range to be found in the sandy soil? . Wang, W J, et al. "Relationships of soil respiration to microbial biomass, substrate availability and clay content." *Soil biology & biochemistry* 35.2 (2003):273-284. ...

Thank you for the suggested reference, we incorporated it (line 64). We also acknowledged the pore size range associated with sandy soils, see lines 335 - 337: *“Sandy soils, such as the ones in this study, are often characterized as having a greater abundance of coarse-sized pores relative to fine pores...”*

F. Clarity and context: lucidity of abstract/summary, appropriateness of abstract, introduction and conclusions

Am reprinting my comments on abstract here.. For this to move forward I would hope for an abstract that explained or postulated why bottom up wetting increased losses from CO₂ and CH₄. Would like the summary to jointly address treatment effects on substrate abundance, and moisture constrains (how does C type inform us about physical access?) - They may be able to exploit differences between CO₂ and CH₄ results to note that

enhanced loss of CH₄ would signal greater amount of area experiencing anaerobic conditions. As presented, results are somewhat intriguing but not informative. to show that wetting direction (simulated precipitation versus groundwater rise) significantly changes pore-scale water C chemistry and core-scale greenhouse gas emissions.

The abstract has been significantly revised – including explanations regarding the effect differential wetting patterns have on fluxes.

In-text comments:

A1: How, state directly

Revised to explicitly state our results and included a rationale for differential wetting patterns from precipitation versus groundwater rise.

A2: Do you mean structurally complex? Or more heterogeneous?

We removed 'complex' and replaced with the actual compounds we were referring to (i.e. lignin, tannin and condensed hydrocarbons). When we used 'complex' in other parts of the manuscript, we made sure to define what we meant by it as well. Thank you for pointing this out.

A3: This is confusing, I think you mean that lipid abundance varies in coarse, carbs in fine and lignin in fine sized pores.. might be clearer if you indicated that the composition of DOC differed in different pore classes and the effects of drought were variable with notable effects on lipids, carbohydrates and lignin coarse, medium and fine pores respectively. Maybe note size here?

We clarified which compounds were altered and in which pores sizes.

A4: How big ?

Revised.

A5: Find this awkward, is this review meant to be comprehensive? For symmetry, then what portion of soils are influenced by permafrost. The theme calls for wetting by saturated versus unsaturated flow?

This has been revised and we did not intended to provide a comprehensive review, just mention that capillary wetting affects more than riparian and wetland soils. Sorry for any confusion.

A6: Be nice to see this to know if the soil used was likely to be representative of soils that have a history of wet dry, freeze thaw, C input regime, and clay mineralogy. Since this is not in press provide some info here to explain why this is likely to be generally true

This has been revised to narrow our focus on wetting effects, and not on changes in C input, freeze-thaw and clay mineralogy, which out outside the scope of this study.

A7: Is it direction of flow that is important or the size of the pores that transmit water?

Thank you for pointing this out. We clarified what we mean by "wetting direction," see lines 73 - 74: "Wetting direction (i.e. from above via precipitation or from below from groundwater rise) produces alternate saturation patterns among different pore-size domains" and lines 82 - 91: "The objective of this research was to develop a molecular understanding of the influence wetting patterns, hereon referred to as wetting direction, and antecedent soil moisture condition have on soil C vulnerability at both the soil core and pore scales"

A8: Direction?

Revised.

A9: So, is this direction of flow or transport path?

Revised. See above response.

A10: blank

A11: So, this paragraph suggests that long-understanding of C pulse following drying is driven by doc released from microbial death – here stocks that had moved deeper into soil are returned to an accessible stock

This paragraph has been sufficiently revised. Discussion of microbial necromass is now included in the manuscript, see lines 267 - 270: "We detected fewer lipids in the loosely-held pore water sampled from drought-conditioned soil, compared to cores maintained at field moisture. This suggests that the

accumulated microbial necromass was rapidly mineralized when drought-induced soils were rewet, leading to higher CO₂ emissions.”

A12: These are very short term studies, what were doc contents in field moist soils?

We included DOC contents in the results and as Extended Data Table 3 for all soils (field moist and drought) in the study.

Reviewer #2 (Remarks to the Author):

This ms reports the impact of two different rewetting sources on CO₂ and CH₄ production of a previously dry riparian soil under laboratory and field conditions. Both, simulated precipitation and groundwater rise affected the CO₂ and CH₄ production as well as the relative abundance of C compounds in soil pore water. The fact that capillary rise of groundwater into dry soil horizons can trigger an increase in microbial processes has been widely overlooked in soil C research. However, the ms has some methodological shortcomings and provides little mechanistic investigations that together allow no generalization of the findings:

(1) Soil cores (0-15 cm depth) were taken from one study site with subtropical climate and then stored at -20{degree sign}C. It seems that the topsoil in the field is not influenced by capillary rise (Fig. 3c, groundwater level varies between -2 and -3m), but the soil at about <-1.5 m. Soil freezing at -20{degree sign}C could have altered the microbial community, which never experienced soil frost, and the amount of available C through killed microbial cells. Taking soil cores (core diameter?) further interrupts hyphae of mycorrhizal and saprotrophic fungi and roots that have the ability to redistribute water in soils along soil water potential gradients.

These soils were collected from The Disney Wilderness Preserve, FL which is located in the Everglades watershed (see Extended Data Figure 2). Soils in this area, even surface soils, experience capillary-led wetting, see lines 93 – 95: *“Intact soil cores were collected from a sandy site located in the Everglades watershed (FL, USA) naturally subject to significant hydrologic variability, including capillary-led wetting⁸”*

The freezing, indeed, has the potential to alter the soil microbial community. It is a requirement of the USDA, however, that these soils be treated thusly to control the distribution of fire ants across the US. Typically, we assume that the act of sampling will shift the community on its own. Therefore, we “conditioned” these soils for 48 hours prior to initiating the experiment, in order to allow the soils to recover from sampling and freezing and acclimate to the lab conditions. Our respiration data indicate that a robust microbial community remained (re-emerged) following this disturbance. As this is a laboratory study, we recognize the artifacts of sampling. However, this study focused on the availability of C to *competent* microorganisms, not necessarily native microorganisms.

(2) The origin of CO₂ and CH₄ is unclear. Isotopic analyses could help to resolve the sources. In particular, changes in the ground water level could lead to a short-term volatilization of gases in addition to changes in microbial gas production. Consumption of CH₄ is another factor controlling the net CH₄ efflux.

Thank you for these points. We provided additional data (e.g. the concentration of organic solutes) that may shed some light on the origin of CO₂ and CH₄. We agree that the study would be strengthened with isotopic analyses; however that is beyond the scope of this study, and this particular site has a mixed C₃/C₄ signal with strong hydrodynamic forces that impede straightforward native isotope analyses.

(3) Drought stress was induced by reduction to ~5% soil moisture. Water potential would have been more appropriate considering variations in soil texture. How is physiological drought stress defined? Is 5% soil

moisture light, moderate or strong drought stress? Small changes in water content can have tremendous effects on the physiological relevant water potential.

This is a great point. We provided moisture retention data in the manuscript (Extended Data Figure 3) and reported approximate water potential information for the moisture contents reported in the methods, see lines 422 - 427: *“Field moist cores were maintained at their original moisture content by weight, ~ 15 % (approx. 1.5 pF). Drought was simulated by placing a set of cores on a dry ceramic pressure plate (1bar Tempe Pressure Cell units, Soil Moisture Equipment Corp. Goleta, CA, USA) and allowing them to evaporate until they reached ~ 5 % moisture by weight (approx. 1.8 pF) in an environmental growth chamber set at 22 °C and 60 % humidity (approx. 30 days) (CMP6050 Conviron, Pembina, ND, USA).”* While we agree that water potential is a more accurate representation of moisture conditions in soils, we would need to perform water retention curves on each core in order to accurately dry down each core to a specific water potential. As such, gravimetric moisture content was the most straight-forward and **robust** measurement available. We acknowledged the importance of water potential in our experimental design by collecting pore water via suction (see our response to #5 below).

We agree that small changes in water content, indeed, have a tremendous effect on the physiologically relevant water potential! We argue that small changes in water location may have tremendous effects, as well. Yet this rarely acknowledged in current literature. This work is our attempt to contribute to our understanding of soils' response to drought at the pore scale.

(4) Oxygen concentration in soil pores is a crucial parameter for the balance of CH₄ and CO₂ production which might have been affected by the direction of wetting. Are data available?

Again, this is a great point. Oxygen concentration is important in determining the ratio between CO₂ and CH₄, but is also difficult to measure directly within specific pore size domains. We believe that it is outside the scope of the current manuscript. We are currently working with modelers to identify oxygen and water contents at the pore scale for a different body of work.

(5) The rationale for sampling of soil pore water by pressure cells at -1.5, -15 and -50 kPa is not clear to me. Were sampled water volumes different for specific pore sizes and treatments? Relative composition of organic solutes (Fig. 2, Ex-Table 2) is less relevant for CO₂ and CH₄ fluxes than total amount of organic solutes.

We included the rationale for suctions used in the methods; see lines 470 – 473: *“Suction strengths were chosen based on preliminary work showing these particular pore water fractions are discrete for these soils²⁵. Using the Kelvin equation²⁸ to estimate the largest water-filled pore diameter, -1.5, -15, and -50 kPa suctions correspond to pore and pore-neck size diameters of approximately 200 μm, 20 μm, and 6 μm³³.”* We also included DOC concentrations and total MS peaks in the results (Figure 3, Extended Data Table 3).

L60-61: 'Groundwater-led wetting affects the vast majority of riparian soils, which hold roughly a third of the global, terrestrial carbon pool.' Do you mean wetlands? Riparian soils, generally defined as soils along rivers and lakes, make a minor contribution to global wetland area. I don't see the global significance of drying/wetting in riparian soils for the global C cycle.

You are correct. Thank you for pointing this out. This has been removed.

Reviewer #3 (Remarks to the Author):

This is a review of the manuscript "Precipitation versus groundwater: rewetting source affects soil carbon loss after drought". The authors used laboratory incubations of intact soil cores to induce antecedent drought conditions (with appropriate controls), and rewetting of cores from either the top down (simulating

precipitation) or the bottom up (simulating groundwater rise). They found that wetting direction and moisture history altered carbon dioxide and methane fluxes from the cores, as well as pore-scale carbon compound distribution. The authors claim that drought and wetting effects on pore-scale C distribution and hydrological connectivity interact to alter core-scale greenhouse gas fluxes with implications at the field scale.

The claims made are novel in that few studies have examined the effects of precipitation and groundwater rise together in the same experiment. It is interesting to see, and makes intuitive sense that precipitation primarily affects carbon dioxide flux and groundwater rise affects methane flux. The data on pore scale carbon-compound distribution are particularly interesting and will probably be the most interesting to other researchers in this area. The results are somewhat difficult to interpret, but the pattern is pretty clear. However, I do think that the authors may have overstated the claim that their data show "that accessibility to microbial degradation is a dominant protect mechanism of C in these soils" (lines 130-131). While their data are consistent with this hypothesis, it is not clear whether changes in compound abundance (lipids in this case) are due to microbial mortality, decomposition, mass transport of microbes and substrate, or even if the isolated lipids are derived from microbial or plant sources. All of these are mentioned in the manuscript, but the data available cannot separate the various fate and transformation mechanisms. I think it would make a better paper if the authors were to back off this claim and discuss in a little more detail each potential process, and how the net effect of these processes might explain the observed patterns.

We clarified our original results and also included additional data to better support our claims regarding microbial accessibility. We also provided more mechanistic, or process-based, descriptions of our pore-scale results and analyses to explain core-scale patterns. We proposed several of the excellent points you brought up in our revised discussion as well. Thank you for these suggestions. Discussion regarding the changes in lipids we see have been included in lines 229 - 231: *"...these more simple forms of C (including lipids) may be more abundant in pore spaces that are larger and more connected - as this is where more decomposers colonize^{19,30}, and where there are probably more recent inputs of litter "* and lines 246 - 248: *"The observed decrease in lipids in loosely-held pore water following antecedent drought may indicate a loss of microbial biomass due to prolonged dry conditions³⁷, as the cellular membrane of microorganisms is dominated by phospholipids^{30,38}"*

We also toned down our claims regarding physical accessibility and provide more detail into both physical and hydrological protection of soil C in the revised manuscript, for example see lines 313 - 320: *"While there was a hydrologic conduit created from capillary-led wetting which connected fine-pore C with decomposers, those decomposers may not have been capable of degrading the 'complex' forms of C associated with finer pores (complex carbon compounds such as lignin or tannin). For example, fungi and other known lignin and tannin degraders may not be able to physically access C located in finer pore domains due to simple pore size exclusion⁴⁶. Thus, competent decomposers need more than a hydrologic conduit in order to access C; they also need physical access, highlighting the importance of both hydrologic and physical accessibility in determining the vulnerability and persistence of C in soil,"* and lines 321 - 327: *"Results from our soil homogenization study further support the above-mentioned explanation of our results. The process of homogenization, by definition, distributed C, resources and decomposers more equally throughout the core, thus removing both the vertical and pore-size controls on C mineralization observed in our results from intact soil cores. Furthermore, despite the sandy nature of these soils, more soluble C was collected from homogenized soil cores compared to intact cores, suggesting that there was a pool of structurally protected C that we were unable to access and identify when we first collected pore water from intact cores"*

I'm also not sure how much the field-based data add to the claims made the authors (paragraph, lines 116-123). I understand that the authors want to tie their core-scale experiments to plot- and field-scale dynamics, but I have some problems with this analysis. First, These data look like they come from a separate field study in that

the methods mention open top chambers placed on the soils to elevated temperatures, yet no temperature treatment was included in the lab incubation experiment. The regression analyses on field carbon dioxide fluxes also include this temperature treatment, possibly confounding the results related to moisture differences. If the analysis were limited to ambient temperature treatments only, this might make for a better comparison between laboratory and field studies. Second, the laboratory results show that precipitation has the strongest effect on carbon dioxide flux and groundwater rise on methane flux, yet the field results, which have carbon dioxide as the response variable, are really only testing the control of precipitation on field-scale carbon flux. It seems that methane flux data at the field scale are really needed to test the combined effects of wetting direction at the field-scale. I think that the manuscript is strong enough to stand on the laboratory results alone and the authors should reconsider how, or even if, these field data should be included.

We agree with you (and the other reviewers) that the results from the field-scale data were unclear and misaligned with our laboratory experiments. We revised the field-scale data and only included ambient plots and while we regret that we don't have methane data for the field-scale data, we think the CO₂ results are valuable. We also make sure to clearly state the limitations of our analysis of the field observations, see lines 359 – 366: *“While this is correlative and observational, and thus not conclusive proof, it is consistent with our laboratory results and supports the idea that soil wetting direction can be a strong control on field-scale CO₂ emissions as well. Given that soil respiration is the second-largest C flux at ecosystem to global scales, and a generally poorly-constrained one⁵³, our results may help generate new hypotheses that could be posed to understand larger-scale C fluxes. Furthermore, the combined analyses of our laboratory experiment with the field experiment suggest that there is the potential to improve current predictions of soil C dynamics by including soil history, and wetting source in future models.”*

Reviewers' comments:

Reviewer #1 (Remarks to the Author):

Major claims about importance of moisture by C cycle interactions are valid. Findings about wetting direction by C mineralization interactions are interesting. Variable effects on CO₂ vs CH₄ in intact cores not fully explored. Effects of wetting direction and pre-treatment on DOC composition interesting and novel, suggestive but not conclusive. Am not convinced that analyses of intact vs homogenous (and previously incubated) soils are correctly done or interpreted. The addition of the field location figure is helpful. Details about soil types and drainage characteristics would be nice. I looked these up and was satisfied that the three locations were similar enough to support comparison being made. The biggest problem in the manuscript is the presentation. It was difficult to read through the confusing and possibly misleading use of terms - presentation of results made one believe that comparisons were more straight forward and conclusive than they actually were.

Comments

Ln 27- confusing.. assuming one reads the abstract first they need to understand what the treatments are. When they refer to 'simulated precipitation versus groundwater rise' one could think they were comparing model-based estimates with actual measured groundwater rise. Later use of language that explains their treatments as 'wetting direction, and antecedent soil moisture conditions' is clearer. Their explanation of intact and homogenized cores is clear (figure summarizing sequence of study and text makes it plain) but some of the discussion (Ln 29-30) and analyses done ignore/obscure the fact that these were actually two studies.

Ln 36, might want to replace 'soil homogenization'- with structure ?

38-39- say how this was done ? Again, someone who hadn't yet read paper might think greenhouse gasses were estimated with simulation modeling.. be explicit

Changes in the composition of OM due to wetting direction and antecedent drought at the pore scale corresponded with core-scale patterns in greenhouse gas (GHG) emissions. (specify CO₂ and CH₄).. Discussion seems to emph CO₂- does not utilize CH₄ results.

40-41Ln

....' precipitation resulted in nearly 5x greater cumulative GHG-C emissions in drought-conditioned soils compared to those at field moisture.' Example of language that is vague confusing.. specify, soil was maintained or preconditioned at field moisture (what is that?) or do you mean field capacity (use of this more specific term would be much clearer!). Field moisture varies depending on weather..

51 Global soils? Understand one is trying to make a case for importance but .. Why not use general term Soils ? To scale to global soils one might choose a more representative soil type or a suite..

Lns 65-66-

This paper and the paper cited address moisture interactions with C cycling and do not really focus on greenhouse gas (GHG) emissions- which many will assume include N₂O particularly since moisture interactions are so important for denitrification. After reading on, one would learn both CO₂ and CH₄ were assessed. Why? How do we interpret differences in the effects of factors on emissions? There are differences but these are not discussed.

Ln 88, subjected to laboratory-induced drought conditions.. add 'prior to wetting' to the end of the sentence.

Ln 91. What is the basis for this hyp asserting sorption/desorption interactions between soil minerals and OM are sole drivers of change in DOC complexity. Might microbial death due to dessication or starvation alter input or utilization?

Lns 104-106

These methods came as a surprise given discussion of treatments..

*So, given the absence of side by side comparison between intact and homogenized cores one needs to note that evaluation of homogenized cores reflects the fact they are also previously leached. Combined effect of history is complex. Less SOC, but disturbance will have released C. Changes in pore size distribution that accompany differences might help. The disappearance of wetting direction effects on CO₂ losses seen in intact cores is interesting. No diff in CH₄.. removal of these results from story might simplify since you don't use.

Ln 172- the following sentence does not acknowledge above..

'Due to the strong and interactive effect soil homogenization had on CO₂ and CH₄ emissions, treatment effects (antecedent drought and wetting direction) on cumulative CO₂-C and CH₄-C were analyzed separately for intact and homogenized cores (**Table 2**).

Above should read. something more like ..* 'due to the fact that these were actually separate experiments? Also, .seems like this text relates to Table 1 not 2?

*Analyses in table 2 seem inappropriate.

Fig 1 The number of points depicting cores does not agree with methods suggesting a total of 16 cores? What was actually done?

Fig 2. By presenting this way without revealing how grinding changed pore volume, and with not stats we cannot tell if anything is significant - suspect not, and again, these are separate studies.

Table 1, letters assigned to CH4 intact cores seem wrong

Lns 835-837- Extended Data Figure 2 Map of field sites at the Disney Wilderness 834 Preserve, FL- nice addition, *Report soils info.. two are mapped as Immokallee one as Smyrna fine sands- similar depths to water table..

Reviewer #2 (Remarks to the Author):

The authors have improved the ms, but there are still some issues that needs to be clarified. One issue is that there was no measurement of the vertical pattern of water potential/water content in the soil cores before rewetting. The procedure of soil drying as described generated an average water content of 5% in the soil core. It is very unlikely that the remaining water was homogenously distributed in the 15 cm soil core as evaporation causes strong drying in the topsoil (open to the atmosphere) and almost no drying in the subsoil (contact to suction plate). Vertical gradients in water content (caused by drying) could have influenced the effect of rewetting on GHG fluxes and the vertical distribution of WSOC compounds. The higher GHG emissions in the precipitation treatment compared to the groundwater rise treatment could be, in part, due to the ongoing microbial activity in the incomplete dried subsoil. Water potential or water contents should have been controlled and adjusted to the same value in both the subsoil and topsoil before rewetting. Homogenous water contents within the soil cores are a precondition to test the central hypothesis.

My second point is that pF 1.8 (=5% gravimetric soil moisture, L. 423) corresponds to field capacity, i.e. no water limitation or drought stress for microbial activity. Either the water retention curve is incorrect or there was no drought stress in the soil cores.

The homogenized soil samples (-1.5 kPa, -50 kPa) had very high WSOC concentrations of 267 – 18001 mg L⁻¹ (Extended Data Table 3) which seems to be unlikely given the small total C content of 0.36-0.60 g C/100g soil (Extended Data Table 4). On the other hand, the intact cores had higher WSOC concentrations at -15 kPa than the homogenized soils. Make sure that you provide correct data.

The diameter of the intact cores (3 cm) are rather small. Does the PVC tube affect the capillary rise of water between soil and inner wall of the tube? Can it be excluded that precipitation water moved downwards along the wall following precipitation? Cores with greater diameter would have reduced the risk of artifacts during rewetting.

Ecosystem respiration (field measurement) does not really contribute to the understanding of the rewetting effect on soil CO₂ efflux. Plants and soil microorganisms may differently respond to precipitation and groundwater rise.

My main point of criticism is that the study covers only the short-term pulse (20 h) of CO₂ and CH₄ fluxes after rewetting. It is not clear if such short-term pulses affect the soil C pool or the GHG budget of the soil on a longer time scale as asserted in the title and conclusion. Repeated drying and wetting cycles with soils from different sites and of different textures would be helpful to verify the conclusion. My impression is that the authors still overstate their findings.

Reviewer #3 (Remarks to the Author):

After reviewing the authors' response to previous reviews, I think they have responded well and the manuscript is much improved. They have managed to narrow the focus and cleared up much of the ambiguity. It appears that the revisions have disrupted the narrative of the manuscript a bit so I encourage the authors to carefully check the text for logical consistency. I found a few typos, but there may be more. It looks like Tables 1 and 2 are mislabeled in the text and presented out of order in the results. Also, lines 178 through 180: Do both sentences refer to intact cores with antecedent drought? These sentences are very confusing as written.

Response to Reviewers, Manuscript 16-14188B

Reviewer #1

Major claims about importance of moisture by C cycle interactions are valid. Findings about wetting direction by C mineralization interactions are interesting. Variable effects on CO₂ vs CH₄ in intact cores not fully explored. Effects of wetting direction and pre-treatment on DOC composition interesting and novel, suggestive but not conclusive. Am not convinced that analyses of intact vs homogenous (and previously incubated) soils are correctly done or interpreted. The addition of the field location figure is helpful. Details about soil types and drainage characteristics would be nice. I looked these up and was satisfied that the three locations were similar enough to support comparison being made. The biggest problem in the manuscript is the presentation. It was difficult to read through the confusing and possibly misleading use of terms - presentation of results made one believe that comparisons were more straight forward and conclusive than they actually were.

We have provided a more detailed description of the scientific rationale and overall objectives for the intact vs. homogenous cores study (see lines 75-77: “...We tested this hypothesis on soil cores that were structurally intact, and for which physical protection may have been a dominant mechanism for C persistence, as well as on soil cores that had been homogenized, so that any effect of physical protection was removed,” and lines 90-96: “To further clarify the importance of physical protection in controlling soil C fluxes, the rewetting experiment was then immediately repeated on the same, previously-intact soil cores after each core was homogenized and repacked (see methods, Supplementary Figure 1). Core-scale CO₂ and CH₄ flux and pore-scale OM composition measured from homogenized cores represents the response from previously protected, or unmeasurable, C that was physically occluded during the first set of the experiment (in intact cores)” and lines 358-361: “In order to measure core-scale fluxes and pore-scale OM chemistry associated with the pool of previously occluded (i.e. physically protected) C, the repacked, homogenized soil cores were immediately rerun through the experiment, starting with the rewetting incubation (Supplementary Figure 1).”). We believe that this will clear up any misgivings regarding the appropriateness of our analyses as it was our intent to use the same cores. Please note that all of the statistical analyses were reviewed with a biostatistician prior to resubmission. At the same time, we provided additional analyses in which the intact and homogenized cores were analyzed separately. Furthermore, we have explored the CH₄ results in more detail (see lines 179-183 in results and lines 259-264 in the discussion: “The increase in cumulative CH₄-C in drought-conditioned cores compared to cores maintained at field moisture only when cores were wetted from below may be driven more so by the functional potential of the microbial to produce CH₄ than the concentration of C. For example, there may be a higher abundance of microorganisms functionally capable of producing CH₄ at the bottom of the core, where O₂ concentrations are less abundant, compared to the top of the core” and provided additional information on the soil types, including drainage, from all the locations sampled/measured (see lines 336-342: “Soils are dominated by sandy textures, and depending on local topographic position show moderate to high levels of SOM accumulation at the surface. The soil is classified as an Immokalee fine sand. The Immokalee series is taxonomically defined as sandy, siliceous, hyperthermic Arenic Alaquods, and is characterized as being poorly drained, friable, strongly acidic, with a weak fine granular structure due to the mixture of organic matter and fine roots,” and lines 469-472: “The soil is classified as a Smyrna sand, of which the series is taxonomically defined as sandy, siliceous, hyperthermic Aeric Alaquods. This series is characterized as poorly drained, friable, strongly acidic, and rapidly permeable at the surface and having a weak coarse granular structure.”). Finally,

we worked on the overall presentation and manuscript narrative to improve readability. Please find responses to your specific concerns below.

Ln 27- confusing.. assuming one reads the abstract first they need to understand what the treatments are. When they refer to ‘simulated precipitation versus groundwater rise’ one could think they were comparing model-based estimates with actual measured groundwater rise. Later use of language that explains their treatments as ‘wetting direction, and antecedent soil moisture conditions’ is clearer. Their explanation of intact and homogenized cores is clear (figure summarizing sequence of study and text makes it plain) but some of the discussion (Ln 29-30) and analyses done ignore/obscure the fact that these were actually two studies.

The original abstract (500+ words) has been revised to meet requirements of Nature Communications; i.e. 150 words and no references. As such, this sentence no longer exists. However, we have provided separated analyses of intact and homogenized cores as per your suggestion.

Ln 36, might want to replace ‘soil homogenization’- with structure ? No longer applicable (see above comment).

38-39- say how this was done ? Again, someone who hadn’t yet read paper might think greenhouse gasses were estimated with simulation modeling.. be explicit. Changes in the composition of OM due to wetting direction and antecedent drought at the pore scale corresponded with core-scale patterns in greenhouse gas (GHG) emissions. (specify CO₂ and CH₄).

No longer applicable for the abstract. However, we made sure to be more specific, especially in regards to which greenhouse gases we refer to throughout the remaining text (i.e. using CO₂ and/or CH₄, or carbonaceous greenhouse gases).

Discussion seems to emph CO₂- does not utilize CH₄ results.

Thank you for pointing this out. We included a more detailed analysis and evaluation of CH₄ results; see lines 179-183 in results 259-264 in the discussion (as reported in first comment above).

40-41Ln...’ precipitation resulted in nearly 5x greater cumulative GHG-C emissions in drought conditioned soils compared to those at field moisture.’ Example of language that is vague confusing.. specify, soil was maintained or preconditioned at field moisture (what is that?) or do you mean field capacity (use of this more specific term would be much clearer!). Field moisture varies depending on weather..

To reduce any confusion with the vague terminology used (e.g. field moisture), we defined this term in the introduction (in addition to the definition that was previously included in the methods). Please refer to lines 81-84: “Sixteen experimental cores were randomly assigned to four factorial treatments of antecedent soil moisture conditions (moisture at time of sampling...)” and lines 366-367: “Field moisture cores were maintained at their original, *in situ* moisture content by weight, ~ 15 %”. We prefer to use language that most accurately represents the moisture content used in our study. Field capacity is a measured value and while it is more ‘specific’ (as you note), it is not accurate for this study. We chose to incubate the soil cores at their *in situ* moisture content (at the time they were sampled) and that is what we reported and defined in the introduction and methods.

Lns 65-66- This paper and the paper cited address moisture interactions with C cycling and do not really focus on greenhouse gas (GHG) emissions- which many will assume include N₂O particularly since moisture interactions are so important for denitrification. After reading on, one would learn both CO₂ and CH₄ were assessed. Why? How do we interpret differences in the effects of factors on emissions? There are differences but these are not discussed.

As stated above, we made sure to be specific and consistent with the gases measured (i.e. not N₂O). We also included additional interpretations for both CO₂ and CH₄ (see lines 179-183 in results 259-264 in the discussion and see response to first comment above).

Ln 88, subjected to laboratory-induced drought conditions.. add 'prior to wetting' to the end of the sentence. **Revised.**

Ln 91. What is the basis for this hyp asserting sorption/desorption interactions between soil minerals and OM are sole drivers of change in DOC complexity. Might microbial death due to dessication or starvation alter input or utilization?

We expanded our hypothesis to include microbial death. Our hypothesis on sorption/desorption is based off of work of our colleagues recently accepted for publication in Nature Communications (NCOMMS-16-23994A) (Newcomb, CJ, NP Qafoku, JW Grate, VL Bailey, JJ De Yoreo. Developing a molecular picture of soil organic matter-mineral interactions by quantifying organo-mineral binding) in addition to the following publications (as cited in the manuscript): Kaiser, et al. *Soil Biology and Biochemistry* **80**, 324-340 (2015) and Aubry, et al. *Water research* **47**, 3109-3119 (2013). All relevant publications supporting our hypotheses are cited in the text.

Lns 104-106 These methods came as a surprise given discussion of treatments...*So, given the absence of side by side comparison between intact and homogenized cores one needs to note that evaluation of homogenized cores reflects the fact they are also previously leached. Combined effect of history is complex. Less SOC, but disturbance will have released

C. Changes in pore size distribution that accompany differences might help. The disappearance of wetting direction effects on CO₂ losses seen in intact cores is interesting. No diff in CH₄..removal of these results from story might simplify since you don't use.

We clarified our objectives with the homogenization part of the study in both the introduction (see lines 90-96 and 358-361, reported in our response to your first comment). It was absolutely our intention to look at the additional C that was not previously 'leached out'. As such, it wasn't an investigation into soil disturbance (compared to non-disturbed cores). The paired study was meant to investigate the structurally protected pool of C that we were unable to detect in the intact cores. **Due to the fact that these are sandy soils, generally lacking in soil structure, we think it is quite compelling that there was a substantial pool of C released with homogenization.**

Pore size distribution has now been included, see lines 160-164: "The effect of soil homogenization on the pore size distribution was minimal (Supplementary Figure 3) with greater frequency (~ 25 %) of pores 150 – 200 um diameter in homogenized cores compared to intact cores (~ 17 %) and a greater overall diversity of pore sizes in intact cores compared to homogenized cores." and Supplementary Figure 3 and Supplementary Media 1 (snapshot of 3D images shown below).

We also provided additional details on the CH₄ results, as mentioned above.

Global soils? Understand one is trying to make a case for importance but .. Why not use general term Soils ? To scale to global soils one might choose a more representative soil type or a suite..

Revised, removed.

Ln 172- the following sentence does not acknowledge above...

‘Due to the strong and interactive effect soil homogenization had on CO₂ and CH₄ emissions, treatment effects (antecedent drought and wetting direction) on cumulative CO₂-C and CH₄-C were analyzed separately for intact and homogenized cores (Table 2). Above should read something more like ..* ‘due to the fact that these were actually separate experiments? Also, seems like this text relates to Table 1 not 2?’

*Analyses in table 2 seem inappropriate.

This has been revised. Please refer to our above response on the analyses of intact and homogenized cores. These were not separate experiments; however, we have separated the analyses as per your comments. Please note, we consulted with a research statistician on the statistical analysis and it was deemed appropriate. Splitting the analysis of pore water chemistry pulled from intact and homogeneous apart does not change the overall results, but we agree that it is easier for the reader to separate all analyses, and not just the core-scale CO₂ and CH₄ data. We also corrected the table orders.

Also, lines 178 through 180: Do both sentences refer to intact cores with antecedent drought?

Revised.

Fig 1 The number of points depicting cores does not agree with methods suggesting a total of 16 cores? What was actually done?

Figure 1, as noted in the text and figure caption, is a principal component analysis (PCA) plot of pore water collected at three difference pore size (suctions) domains from those 16

cores, as such each figure section should contain no more than 32 points. In fact, the number of points per plot is actually reported in the figure caption: “Blue shaded points and solid-shaded 90 % confidence interval ellipse represent pore water collected from soil cores subjected to field moisture and precipitation-led rewetting conditions ($n = 8, 8$ and 7 for $-1.5, -15$ and -50 kPa pore water fractions, respectively), whereas open blue circles and pattern-filled confidence intervals represent field moisture core rewet from below to simulate groundwater rise ($n = 8, 8, 7$). antecedent drought and rewet via simulated precipitation ($n = 8, 6, 7$), whereas ... drought-induced samples rewet via simulated groundwater rise ($n = 7, 6, 5$).”

Fig 2. By presenting this way without revealing how grinding changed pore volume, and with not stats we cannot tell if anything is significant - suspect not, and again, these are separate studies.

Soil homogenization (not grinding) would alter the specific pore structures present when the core was intact, but it would not change particle size distribution. Porosity is included as Supplementary Table 4 for both intact and homogenized cores. In addition, pore size distribution via X-ray tomography has now been included in the follow ways: (1) in text, see lines 160-164, (2) as Supplementary Figure 3, and (3) as Supplementary Media 1.

The statistics for Figure 2 are included as Table 2. We revised table 2 by separating the analysis for intact and homogenous cores as per your suggestion. We found that a figure coupled to a statistical table (in addition to a supplemental table that lists the means and standard errors for the data presented in Figure 2) was more visually appealing and understandable to readers than just listing the information in table-form. We did attempt to revise Figure 2 in order to highlight the statistical effects of the treatments and pore water fractions and even created some new figures/analyses (such as a heat map as suggested by our statistical advisors), but after internal review we chose to keep the original figure as it was more appropriate display of 216 different values and better fit for a general audience.

Table 1, letters assigned to CH4 intact cores seem wrong

The letters are correct. We revised the format and units from scientific notation depicting mg of CH₄-C to µg CH₄-C with 4 significant digits. I can see how one might have missed the difference when such small values were written in scientific notation (e.g. $1.03e^{-4}$ vs. $1.17e^{-5}$).

Lns 835-837- Extended Data Figure 2 Map of field sites at the Disney Wilderness 834 Preserve, FL- nice addition, *Report soils info.. two are mapped as Immokallee one as Smyrna fine sands– similar depths to water table..

Thank you for bringing this up. There were two locations where soils were studied; one where we collected the soils for the laboratory experiment and the other where the field experiment took place and we forgot to include the soil type of the field study in our earlier drafts. This is now revised in both the methods (lines 336-342, 469-472) and is also included on Supplementary Figure 2.

Reviewer #2

The authors have improved the ms, but there are still some issues that needs to be clarified. One issue is that there was no measurement of the vertical pattern of water potential/water content in the soil cores before rewetting. The procedure of soil drying as described generated an average water content of 5% in the soil core. It is very unlikely that the remaining water was homogenously distributed in the 15 cm soil core as evaporation causes strong drying in the topsoil (open to the atmosphere) and almost no drying in the subsoil (contact to suction plate). Vertical gradients in water content (caused by drying) could have influenced the effect of rewetting on GHG fluxes and the vertical distribution of WSOC compounds. The higher GHG emissions in the precipitation treatment compared to the groundwater rise treatment could be, in part, due to the ongoing microbial activity in the incomplete dried subsoil. Water potential or water contents should have been controlled and adjusted to the same value in both the subsoil and topsoil before rewetting. Homogenous water contents within the soil cores are a precondition to test the central hypothesis.

We understand your concerns regarding the heterogeneous nature of soils and our ability to control water potential through the soil cores. We want to assure you that the drought-conditioned soils were dried throughout the soil core. Within < 4-5 days all cores had reached a constant weight and had visibly dry soil at both the bottom and the top of the soil core (and were then further incubated for an additional 28-30 days). The suction plate (as you noted) was a dry ceramic plate and actually did a wonderful job of wicking some of the moisture away from the bottom of the soil core. However, we also confirmed that soil was dry throughout the soil core via destructive analysis on a few cores immediately following our laboratory drought. All soil, even soil in the middle and bottom of the core, was dry. As such, we do not believe that the higher GHG emissions in the precipitation treatment compared to the groundwater rise treatment was due to ongoing microbial activity from subsoil that was incompletely dried.

However, there is truly no way to ensure that an intact soil core has absolute homogeneity for water content/water potential, especially at pore-scale resolution. The finest pores will often retain water (in films) even in drought-stressed soils and we expect that this was true in our cores, and as our motivations were to represent natural soil conditions and responses (albeit in the laboratory), our central hypothesis was not dependent on homogeneity for many properties, not just moisture content.

My second point is that pF 1.8 (=5% gravimetric soil moisture, L. 423) corresponds to field capacity, i.e. no water limitation or drought stress for microbial activity. Either the water retention curve is incorrect or there was no drought stress in the soil cores.

Thank you for pointing this out. We are working with Decagon to identify errors in our water retention curve. See above response regarding dryness of drought-conditioned cores.

The homogenized soil samples (-1.5 kPa, -50 kPa) had very high WSOC concentrations of 267 – 18001 mg L⁻¹ (Extended Data Table 3) which seems to be unlikely given the small total C content of 0.36-0.60 g C/100g soil (Extended Data Table 4). On the other hand, the intact cores

had higher WSOC concentrations at -15 kPa than the homogenized soils. Make sure that you provide correct data.

The high concentrations of WSOC in homogenized samples may appear “unlikely” given the low C content of soils, however, it was not surprising to us after seeing the pore water that emerged from homogenized cores (see picture below). The darkest water in the picture represents the more tightly held pore water fraction (- 50 kPa) and appears to be quite rich in C. The large error and heterogeneity observed for WSOC in the pore water samples are [unfortunately] correct data – but we did notice a labelling error and have revised it (thank you!). You are welcome to review the raw data, which is publically available at:

<https://doi.org/10.6084/m9.figshare.5031362.v1>

However, we provided an additional cautionary note in the methods clarifying the high variation in the methods, see lines 152 -153: “Due to technical limitations with low volume samples (see Methods), the total concentration of organic carbon and nitrogen in pore waters was highly variable (Supplementary Table 3)” and 432-434: “For low volume pore water samples, we prioritized characterizing the molecular composition of C in pore water rather than measuring the concentration of water soluble N (WSN) and organic C (WSOC), due to technical limitations in measuring low volume samples for WSN and WSOC” Additionally, the total count of FT-ICR-MS peaks is included in the results as a surrogate for the richness of OM in the pore water (lines 144 – 151: “An indicator of OM richness, i.e. the total number of C features (m/z peaks) identified using FT-ICR-MS, increased with soil homogenization ($p = 0.001$), antecedent drought ($p = 0.011$) and simulated groundwater rise ($p = 0.034$) (Figure 3). Effective pore size domain did not have an effect on total peaks identified, except in intermediately held pore waters collected from soil cores subjected to drought ($p = 0.049$), where peaks increased. When intact and homogenized cores were analyzed separately, antecedent drought resulted in more FT-ICR-MS peaks in all pore water from intact cores ($p = 0.0497$), whereas in homogenized cores antecedent drought increased the number of peaks only in intermediately held pore water (-15 kPa) ($p = 0.0377$).”

We argue that there is indeed a physically protected pool of C that was previously undetectable (i.e. in intact cores), and that spatial location in the pore matrix (in regards to pore size) matters when it comes to that protected pool. As you note, in -50 kPa pore water (finest pores) there was more C from homogenized cores than intact (as it was previously occluded in intact cores), whereas at lower suctions (i.e. the coarsest pores) it was easy to access the C from intact cores (it was not occluded) and we probably pulled out the majority of C available from that pore size domain when cores were intact, resulting in little C left over in the coarse pores (i.e. easily accessible) from homogenized cores.

The diameter of the intact cores (3 cm) are rather small. Does the PVC tube affect the capillary rise of water between soil and inner wall of the tube? Can it be excluded that precipitation water moved downwards along the wall following precipitation? Cores with greater diameter would have reduced the risk of artifacts during rewetting.

There is always the risk of edge effects when studying intact soil cores. We are confident, however, that our core size was appropriate for our experiment. We have both measured and modeled water imbibition in cores of this size and found that water imbibition fits the pore-flow predictions very well, for flow from below (see Figure 11, Yang et al 2014, doi:10.2136/sssaj2013.05.0190).

Ecosystem respiration (field measurement) does not really contribute to the understanding of the rewetting effect on soil CO₂ efflux. Plants and soil microorganisms may differently respond to precipitation and groundwater rise.

We provided additional information regarding the vegetation present in the chambers that measured soil and groundcover respiration, see lines 474-475: “CO₂ measured from chambers may also include respiration from existing understory vegetation consists; wiregrass (*Aristida stricta*) and palmetto (*Serenoa repens*)” We agree that plant and soil microorganisms would respond differently to precipitation or groundwater rise. We also agree that the study would be cleaner if the chamber were just soil flux (of both CO₂ and CH₄), but we do not directly compare the lab with the field results and we make sure to point out the caveats with including the field study (see lines 307-311: “We found that the *in situ* CO₂ flux responded to both precipitation events and to fluctuations in groundwater level. While this is correlative and observational, and thus not conclusive proof, it is consistent with our laboratory results and supports the idea that soil wetting direction can be a strong control on field-scale CO₂ emissions as well”). However, the fact that our laboratory study can explain some of the variation in the natural system (which has plants in it) in response to precipitation wetting is just a first step toward scaling and compelling enough to report.

My main point of criticism is that the study covers only the short-term pulse (20 h) of CO₂ and CH₄ fluxes after rewetting. It is not clear if such short-term pulses affect the soil C pool or the GHG budget of the soil on a longer time scale as asserted in the title and conclusion. Repeated drying and wetting cycles with soils from different sites and of different textures would be helpful to verify the conclusion. My impression is that the authors still overstate their findings.

We agree that the study would be strengthened with additional soil types and textures. The high peaks in CO₂ or CH₄ that are observed during or immediately following a wetting event influence the amount of cumulative C emitted overall (in the short-term and long-term). We believe this is important to capture and yet, it is rarely reported in the literature. Investigating repeated drying and wetting cycles, or budgets on a longer time scale is beyond the scope of the research we were funded to do. As such, this was highlighted in our study justification and objectives (see lines 58-61: “These responses are often rapid and short-lived, occurring within 24 – 48 hours²⁴⁻²⁸; however, short-term responses can result in significant C losses from rewetting, because such “hot moments” can comprise a nontrivial fraction of the landscape-scale or annual flux budget,” and lines 64-65: “Given the immediacy of the microbial responses to rewetting, per the Birch effect, we focused our measurements on the 20 hours immediately following rewetting.”) and were careful not to overstate our results (see lines 324-325: “Our laboratory experiment cannot accurately represent *in situ* phenomena, but is consistent with our field-scale observations, suggesting that precipitation and groundwater fluctuations may interact to destabilize soil C at the field scale.”). **We constrained our interpretations to the specific results we observed, which do indeed show that there was a significant interaction between**

rewetting direction and antecedent soil moisture condition in both the core- and pore-scale results. Finally, we revised our title to better reflect these results.

Reviewer #3

After reviewing the authors' response to previous reviews, I think they have responded well and the manuscript is much improved. They have managed to narrow the focus and cleared up much of the ambiguity. It appears that the revisions have disrupted the narrative of the manuscript a bit so I encourage the authors to carefully check the text for logical consistency. I found a few typos, but there may be more. It looks like Tables 1 and 2 are mislabeled in the text and presented out of order in the results. Also, lines 178 through 180: Do both sentences refer to intact cores with antecedent drought? These sentences are very confusing as written.

Thank you for your time. We revised the manuscript narrative and polished the typos and mislabeling. Tables 1 and 2 have been revised to be better aligned in the results and the sentence in question (lines 178 through 180) has also been revised (see new lines 169-172: *“When intact and homogenized cores were analyzed separately, there was a significant interactive effect of rewetting direction and antecedent drought on cumulative CO₂-C and CH₄-C in intact cores, whereas there were no significant treatment effects on cumulative CO₂-C and CH₄-C in homogenized cores”*).

REVIEWERS' COMMENTS:

Reviewer #4 (Remarks to the Author):

The main point of this article is to verify the major constraint of C decomposition is not only from physical protection, but also from wetting and drying effects of hydraulic connectivity. The background and rationale are meaningful and relatively well written. However, the way how discussion section is written still needs to be revised. It would be better to have sub-sections within the discussion section for the readers to follow the flow of the story. As the authors have several hypotheses, the discussion could be better developed following each hypothesis.

Novelty of this paper is worthwhile to be published if authors pay more attention to the way of presentation in the discussion section.

Response to Reviewer #4
NCOMMS 14188C

Reviewer 4 The main point of this article is to verify the major constraint of C decomposition is not only from physical protection, but also from wetting and drying effects of hydraulic connectivity. The background and rationale are meaningful and relatively well written. However, the way how discussion section is written still needs to be revised. It would be better to have sub-sections within the discussion section for the readers to follow the flow of the story. As the authors have several hypotheses, the discussion could be better developed following each hypothesis. Novelty of this paper is worthwhile to be published if authors pay more attention to the way of presentation in the discussion section.

Thank you for your review. We revisited each of our hypotheses and provided additional development following each hypothesis in the discussion section.

- Line 231 – 234, (p. 10) “*This is consistent with our hypothesis that the abundance of complex C compounds (such as lignin, tannin, and condensed hydrocarbons) would increase in pore water collected from soil subjected to antecedent drought. The relative increase in complex C compounds may be due to a negative enrichment from preferential degradation of other compounds (such as lipids, from above), or due to the physio-chemical relationship between ionic strength and the sorption of C to mineral surfaces.*”
- Line 242- 248, (p. 11) “*As we hypothesized, we detected fewer lipids in the loosely-held pore water sampled from drought-conditioned soil compared to cores maintained at field moisture, suggesting accumulated microbial residues was rapidly mineralized when drought-conditioned soils were rewet, leading to higher CO₂ and CH₄ emissions.*”

We are unable to follow your suggestion about sub-section headers in the discussion section as per the formatting requirements of the journal.